# Is Knowledge Power?
# On the (Im)possibility of Learning from Strategic Interactions

**Nivasini Ananthakrishnan**[1], **Nika Haghtalab**[1], **Chara Podimata**[2], and **Kunhe Yang**[1]

[1]UC Berkeley, {nivasini,nika,kunheyang}@berkeley.edu
[2]MIT & Archimedes AI, podimata@mit.edu

## Abstract

When learning in strategic environments, a key question is whether agents can overcome uncertainty about their preferences to achieve outcomes they could have achieved absent any uncertainty. Can they do this solely through interactions with each other? We focus this question on the ability of agents to attain the value of their Stackelberg optimal strategy and study the impact of information asymmetry. We study repeated interactions in fully strategic environments where players' actions are decided based on learning algorithms that take into account their observed histories and knowledge of the game. We study the pure Nash equilibria (PNE) of a meta-game where players choose these algorithms as their actions. We demonstrate that if one player has perfect knowledge about the game, then any initial informational gap persists. That is, while there is always a PNE in which the informed agent achieves her Stackelberg value, there is a game where no PNE of the meta-game allows the partially informed player to achieve her Stackelberg value. On the other hand, if both players start with some uncertainty about the game, the quality of information alone does not determine which agent can achieve her Stackelberg value. In this case, the concept of information asymmetry becomes nuanced and depends on the game's structure. Overall, our findings suggest that repeated strategic interactions alone cannot facilitate learning effectively enough to earn an uninformed player her Stackelberg value.

## 1 Introduction

Learning to act in strategic environments is fundamental to the study of decision making under uncertainty in a wide range of applications, such as security, economic policy, and market design (e.g., [31, 8, 19]). In these environments, acting and learning are intimately connected: agents' actions and the reactions they elicit generate payoffs, and help clarify the latent preferences of other agents. A central question is whether, through repeated interactions alone, agents (aka *players*) can overcome uncertainty about each other's preferences in order to achieve outcomes they could have achieved in the absence of uncertainty. An extensive line of work on *learning in Stackelberg Games* [5, 8, 31, 35] has focused on answering this question for achieving the *Stackelberg value*, which is the optimal payoff a player guarantees herself when assuming other players will best respond to her actions. While a player who hopes to attain her Stackelberg value in a one-shot game must know the game (i.e., know the utilities of all players), this line of work asks whether a player who is a-priori uninformed can overcome her lack of knowledge and attain her Stackelberg value through repeated interactions with other players.

By and large, existing works have studied this question by constructing learning algorithms for uninformed players that attain their Stackelberg value, through repeated interactions with other players who myopically best respond. While these results are encouraging for learning about the

38th Conference on Neural Information Processing Systems (NeurIPS 2024).

preferences of well-behaved best responding agents[1], they do not provide clear evidence of the ability of uninformed players to learn from strategic interactions alone. Indeed, the two players' different attitudes towards their outcomes — namely, one player planning a long-term strategy to maximize her long-term payoff while the other player responding without considering the impact of her actions on her long-term payoff — confounds the overall impact uncertainty may have on how well players learn from strategic interactions; leaving one to wonder whether it was the lack of rational long-term planning on the part of one agent or some genius of the learning algorithm employed by the other agent that enabled her to learn from strategic interactions.

In this paper, we revisit the problem of learning in strategic environments with a renewed focus on the impact of information asymmetry between two equally rational players who aim to maximize their total payoff. We ask again: *can an uninformed player learn to attain the value of her Stackelberg outcome, through repeated interactions alone?* In contrast to the aforementioned results, our findings largely imply that *strategic interactions alone cannot facilitate learning effectively enough to earn an uninformed player the value of her optimal strategy.*

**Our Model and Contributions.** To study the impact of informativeness on the ability of players to gain the payoff of their Stackelberg outcome, we study repeated interactions between two rational agents playing repeatedly a one-shot game $G \sim \mathcal{D}$. While the players know $\mathcal{D}$, they may not know the realized game $G$ except perhaps through signals of differing precision about $G$. For example, one player may know $G$ and another may have access to a signal that reveals $G$ with probability $0.5$ and is uninformative (i.e., independently drawn from $\mathcal{D}$) otherwise. Each player deploys an algorithm, which specifies her actions at every round, given all that the player has observed so far (e.g., history of actions and/or utilities experienced) and the information she possesses about $G$. We consider a meta-game where players' actions are algorithms that specify agent's strategies in $T$ rounds of interactions and study pairs of algorithms that form pure Nash Equilibria in the meta-game. We use the overall utility attained by pairs of algorithms that form pure Nash Equilibria to draw a clear separation between the informed and uninformed players' ability to attain the value of their Stackelberg optimal strategies.

In the following, we use $\text{StackVal}_i(G)$ to refer to player $i$'s value of her optimal Stackelberg strategy in game $G$. We summarize our results as follows.

- In Section 3, we study the *full information asymmetry* when player 1 ($\text{P}_1$) knows the realized game $G$ and player 2 ($\text{P}_2$) only has partial information based on an imperfect signal. We show a full separation in the achievable utilities. In particular, we show in Theorem 3.1 that for every distribution $\mathcal{D}$ and realized game $G$, there is a pure Nash Equilibrium (PNE) in the meta game induced between the algorithms' of the two players for which $\text{P}_1$ achieves her Stackelberg value, i.e., $\text{StackVal}_1(G)$. On the other hand, Theorem 3.2 gives a distribution $\mathcal{D}$ such that no PNE of the meta game allows $\text{P}_2$ to achieve $\mathbb{E}_{G \sim \mathcal{D}}[\text{StackVal}_2(G)]$. In other words, for some realized game $G$ and all PNE of the meta-game, $\text{P}_2$ cannot achieve the value of her optimal Stackelberg strategy for $G$.

  Taken together, Theorem 3.1 and Theorem 3.2 establish that learning through interactions alone is not sufficient to allow an uninformed player (in this case, $\text{P}_2$) to attain the value of her optimal Stackelberg strategy. Does this mean that $\text{P}_2$ in unable to *learn* the game matrix through interactions that form a PNE in the meta-game? This is not necessarily the case (see Observation A.4) as indeed $\text{P}_2$ may be able to learn the underlying game $G$ eventually. What our results imply is that in every PNE, either $\text{P}_2$ never learns the game $G$ sufficiently well, or she has enough information to identify $G$ but the stability condition for her algorithm to be in a PNE does not allow her to extract the value of her optimal Stackelberg strategy. This points to the limitations on what agents can achieve if their only source of learning is through repeated interactions.

- In Section 4, we study a setting where neither player fully knows the realized game $G$. Interestingly, a separation need not hold in this case. In particular, there are distributions $\mathcal{D}$ where the player with a less informative signal about $G$ is able to extract her benchmark $\mathbb{E}_{G \sim \mathcal{D}}[\text{StackVal}(G)]$ while the player with the more informative signal cannot achieve her corresponding benchmark. This occurs when the less-informed player is able to learn the identity of $G$ more efficiently,

---

[1]Beyond myopic best-responding, several other types of algorithms for the responding agent have been considered, such as gradient descent [21, 39], no-regret [25, 10], no-swap regret [10], and responding to calibrated forecasts [28]; none of these focuses on how the other player's learning dynamics may be affected as the result of the second player's actions.

possibly due to the structure of $\mathcal{D}$. This is perhaps not surprising, given that a less-informed player can become more informed or even perfectly aware of the realized game faster, while the player who started the meta-game with a more informative signal continues to remain only partially informed.

**On the possibility and impossibility of learning from strategic interactions.** We view our work as providing a different lens on studying learnability in the presence of strategic interactions that also elucidates the context and subtleties of a vast line of prior work in this space. By and large, prior work in this space [31, 8, 5, 35, 11, 25, 22, 15, 38, 28, 10] has attempted to establish the following message: "An uninformed player can always learn to achieve (even surpass) its Stackelberg value through repeated strategic interactions alone". At a high level, our work demonstrates the opposite, that "In some cases, an uninformed player cannot learn, through repeated interactions alone, to achieve its Stackelberg value". Of course, these messages, while both technically correct, are contrary to each other. So, what accounts for this difference?

One of our takeaways is that prior work's findings (that an uninformed can always overcome her informational disadvantage through repeated strategic interactions) heavily hinges on the lack of rationality of at least one of the agents in those strategic interactions. That is, the dynamics studied in prior work involve pairs of agent algorithms that are not best-responses to each other. On the other hand, our work shows that the inherent uncertainty about the game — or more precisely, the information asymmetry between two equally rational agents — can persists throughout repeated interactions and makes it impossible for an uninformed agent to overcome her informational disadvantage.

The processes of learning and acting based on the learned knowledge are naturally intertwined when dealing with uncertainty in strategic environments. Our work implies that it is precisely because of their intertwined nature that an uninformed agent cannot overcome her informational disadvantage from strategic interactions alone. That is, information disadvantage between a pair of rational agents persists for one of two reasons: Either actions taken by the agents' algorithms do not reveal enough information to identify the game at play, or if they do, the less-informed agents use of the elicited information would have lead the informed agent to deviate to an algorithm that barred her from learning in the first place.

## 1.1 Related Works

**Algorithms and benchmarks for repeated principal-agent interactions.** There is a vast literature investigating online algorithms and benchmarks in repeated games with agents under various behavioral models such as: 1) myopically best-responding [31, 5, 8, 38]; 2) optimizing time-discounted utilities [27, 29, 2, 1]; 3) employing no-regret [9, 17, 22, 25, 10, 26, 18], no-swap-regret [17, 32, 10], no counterfactual-internal-regret algorithms [11, 15], or online calibrated forecasting algorithms [28].

Given a particular model of the agent, what is the optimal algorithm to employ? This has been studied in both the complete and incomplete information setting. In the complete information setting, the static algorithm of playing the optimal Stackelberg strategy is shown to be optimal against no-swap-regret agents [17, 28]. But it is not necessarily optimal against general no-regret algorithms, including common algorithms such as EXP3 [9, 17, 26, 36]. Additionally, it is not optimal against no-swap-regret agents in Bayesian games where agents have hidden information but is optimal if agents satisfy a stronger notion called no-polytope-swap-regret [32].

**Long-term rationality of agents in the meta-game.** Instead of modeling agents as no-regret learners, another line of research treats the repeated game as a *meta-game* in which players' actions are their choice of *algorithms*. Towards understanding the PNE of this meta-game, Brown et al. [10] show that no pair of no-swap-regret algorithms can form a PNE unless the stage game has a PNE. Previous work discussed above on optimally responding to no-regret agents also has implications on the meta-game's PNE such as (1) no-swap-regret algorithms are supported in a meta-game PNE for all games $G$ [17], and (2) there are games where no meta-game PNE contains certain common regret-minimizing algorithms such as EXP3 [10]. We discuss these implications in Appendix C.

Kolumbus and Nisan [30] study a meta-game where players are restricted to choose no-regret algorithms but have the option to manipulate their private information. They show that non-truthful PNE exists in multiple classes of games. Besides PNE, some previous works also study Stackelberg strategies of meta-games [14, 40]. Recently, Arunachaleswaran et al. [3] study the Pareto optimality relative to all possible games instead of exact optimality in a particular game.

**Information asymmetry in repeated games.** The final line of related work is the substantial literature on information asymmetry and repeated interactions including classical work by Aumann et al. [4]. They also study repeated games between a player knowing the game and one who does not. For zero-sum games, they show that all PNE of the meta-game yields the same utility to the informed player and this utility can be higher than the informed player's one-shot utility. The higher utility is due to the informed player's ability to shape the learned beliefs of the uninformed player and this power of information is also shown in a recent line of work on follower deception in Stackelberg games [24, 33, 23, 6, 13, 12].

## 2 Model and Preliminaries

We study games between two players, referred to as $P_1$ and $P_2$. Wlog, we assume that $P_1$ is generally *more informed* than $P_2$ (to be defined formally below). Although we focus on *repeated* games, we first provide the setting for one-shot games and then build upon it for repeated games.

**Bayesian Games.** A game $G$ is a tuple $(\mathcal{A}_1, \mathcal{A}_2, U_1, U_2)$, where $\mathcal{A}_i$ is $P_i$'s discrete action space, and $U_i : \mathcal{A}_1 \times \mathcal{A}_2 \to \mathbb{R}$ is $P_i$'s utility function ($i \in \{1, 2\}$). A Bayesian game is described by a family of games $\mathcal{G}$ and a distribution $\mathcal{D} \in \Delta(\mathcal{G})$ over games in this family, where $\mathcal{G}$ consists of games sharing the same action space. When $P_1$ plays action $x \in \mathcal{A}_1$ and $P_2$ plays action $y \in \mathcal{A}_2$, then they receive utilities $U_1(x, y)$ and $U_2(x, y)$ respectively. We sometimes overload notation and write $U_1(x, y; G), U_2(x, y; G)$ to denote that the utilities of the two players come from a particular game instance $G$. Instead of pure strategies (i.e., playing discrete actions), the players can also choose to play *mixed* strategies $\mathbf{x} \in \Delta(\mathcal{A}_1)$ and $\mathbf{y} \in \Delta(\mathcal{A}_2)$ for players 1 and 2 respectively. To simplify notation, we sometimes write $U_i(\mathbf{x}, \mathbf{y})$ in place of $\mathbb{E}_{x \sim \mathbf{x}, y \sim \mathbf{y}}[U_i(x, y)]$ for $P_i$'s utility. Unless specified otherwise, we assume that the players are moving *simultaneously* and they both know the prior distribution $\mathcal{D}$. We also assume that every game $G$ in the support of $\mathcal{D}$ has no *weakly dominated action* for either player. An action $x_0 \in \mathcal{A}_1$ is weakly dominated for $P_1$ in $G$ if there exists $\mathbf{x} \in \Delta(\mathcal{A}_1 \setminus \{x_0\})$ s.t., $U_1(\mathbf{x}, y; G) \geq U_1(x_0, y; G)$ for every $y \in \mathcal{A}_2$. The weakly dominance property of actions $y_0 \in \mathcal{A}_2$ is defined symmetrically for $P_2$.

**Optimistic Stackelberg Value.** The optimistic Stackelberg value of a game $G$ for $P_1$, denoted with $\text{StackVal}_1(G)$, is the optimal value of the following optimization problem:

$$\text{StackVal}_1(G) \triangleq \max_{\mathbf{x}^\star \in \Delta(\mathcal{A}_1)} \max_{y \in \text{BR}_2(\mathbf{x}^\star; G)} U_1(\mathbf{x}^\star, y),$$

where $\text{BR}_2(\mathbf{x}^\star; G) \triangleq \text{argmax}_{y \in \mathcal{A}_2} U_2(\mathbf{x}^\star, y)$ indicates $P_2$'s set of best responses to $\mathbf{x}^\star$. When there are multiple actions in $\text{BR}_2(\mathbf{x}^\star; G)$, ties are broken *optimistically* in favor of $P_1$. We use $(\mathbf{x}^\star(G), y(\mathbf{x}^\star; G))$ to denote the pair of strategies that achieves the value $\text{StackVal}_1(G)$. For $P_2$, the optimistic Stackelberg value $\text{StackVal}_2(G)$ and $(x(\mathbf{y}^\star; G), \mathbf{y}^\star(G))$ are defined symmetrically. Finally, we define $\text{StackVal}_i(\mathcal{D}) \triangleq \mathbb{E}_{G \sim \mathcal{D}}[\text{StackVal}_i(G)]$ to be the *expected* optimistic Stackelberg value for $P_i$ under the prior distribution $\mathcal{D}$ ($i \in \{1, 2\}$).

**Game Information.** We assume that both players know the prior $\mathcal{D}$. After the game $G \sim \mathcal{D}$ is realized, each player $P_i$ also receives additional information about the realization of $G$, which is characterized by a signal $s_i \in \mathcal{G}$. We assume that nature generates both signals $s_1, s_2$ independently and with potentially different precision levels $p_1, p_2 \in [0, 1]$, and each player can only observe their own signal. Fixing a precision $p_i$, $s_i$ perfectly reveals the true game $G$ with probability $p_i$, and with probability $1 - p_i$, it provides an independent draw from the prior distribution $\mathcal{D}$. Formally, the conditional distribution of $s_i$ given $G$ is defined as follows:

$$\forall G, s_i \in \mathcal{G}, \quad \varphi_{p_i}(s_i \mid G) = p_i \cdot \mathbb{1}\{s_i = G\} + (1 - p_i) \cdot \mathcal{D}(s_i).$$

While each player can only observe their own signal $s_i$, we assume that the distributions generating both signals are common knowledge, i.e., both players know $p_1$ and $p_2$. Note that when $p_i = 1$, the signal $s_i$ perfectly correlates with the realization $G$, in which case we say that $P_i$ is *fully-informed* or have *perfect knowledge* about which game is being played. On the other extreme, if $p_i = 0$, then the signal $s_i$ reveals no additional information compared to the prior distribution $\mathcal{D}$. In this case, we call $P_i$ *uninformed*. Throughout this paper, we focus on settings with *information asymmetry* where we always assume $P_1$ is more informed than $P_2$, i.e., $p_1 > p_2$.

**Repeated Games.** In this paper, we focus on repeated (Bayesian) games. Initially, nature draws a game $G \in \mathcal{G}$ from prior $\mathcal{D}$. The game is then fixed and repeated for $T$ rounds. At each round $t \in [T]$, $P_1$ and $P_2$ play strategies $\mathbf{x}^t, \mathbf{y}^t$ and obtain utilities $U_1(\mathbf{x}^t, \mathbf{y}^t; G), U_2(\mathbf{x}^t, \mathbf{y}^t; G)$ respectively. We call $G$ the *stage game* of the repeated interaction.

Without loss of generality, we use *algorithms* to describe both players' adaptive strategies in the repeated game. For $i \in \{1, 2\}$, we use $\pi_i$ to denote the algorithm used by $P_i$, which is a sequence of mappings $(\pi_i^t)_{t \in [T]}$ that at each round maps from player $i$'s information about the game and historical observations to the distribution of mixed strategies from which the next strategy is drawn. Specifically, for each round $t$, the mapping is defined as $\pi_i^t : (s_i; H_i^{1:t-1}) \mapsto \Delta(\mathbf{x}_t)$, where $s_i$ is the signal received by $P_i$ about the realization of $G$, and $H_i^r$ is the feedback that $P_i$ observed at round $r$ ($r \in [t-1]$). When both players observe each other's realized strategies as well as their own (but not the other's) realized utilities, we have $H_i^r = (\mathbf{x}^r, \mathbf{y}^r, U_i(\mathbf{x}^r, \mathbf{y}^r; G))$. We call this the *full-information feedback* setting. We also consider the *bandit feedback* setting, where the players do not observe the strategies of their opponent, i.e., $H_1^r = (\mathbf{x}^r, U_1(\mathbf{x}^r, \mathbf{y}^r; G))$ and $H_2^r = (\mathbf{y}^r, U_2(\mathbf{x}^r, \mathbf{y}^r; G))$.

**Trajectories and expected utilities.** Consider a fixed pair of algorithms $(\pi_1, \pi_2)$. Under every realization of $(G, s_1, s_2)$, algorithms $(\pi_1, \pi_2)$ induce a distribution over trajectories of mixed strategy pairs of length $T$, which we denote with $(\mathbf{x}^t, \mathbf{y}^t)_{t \in [T]} \sim \mathcal{T}^T(\pi_1, \pi_2; G, s_1, s_2) \in \Delta(\Delta(\mathcal{A}_1)^T \times \Delta(\mathcal{A}_2)^T)$. In particular, the signals $s_i$ ($i \in \{1, 2\}$) are inputs of $\pi_i$ that specify $P_i$'s behavior upon receiving certain feedbacks, whereas $G$ influences $P_i$'s observed utilities, which is part of the feedback and indirectly influences $P_i$'s strategies of the next round.[2] We also use $\mathcal{T}^T(\pi_1, \pi_2; G)$ to denote the mixture of $\mathcal{T}^T(\pi_1, \pi_2; G, s_1, s_2)$ as $s_i \sim \varphi_{p_i}(\cdot \mid G)$ for $i \in \{1, 2\}$.

When the realized game is $G$ and players use algorithms $(\pi_1, \pi_2)$ with time horizon $T$, the *expected average utility* of $P_i$ under $G$, denoted as $\bar{U}_i(\pi_1, \pi_2; G)$, can be expressed as

$$\bar{U}_i^T(\pi_1, \pi_2; G) \triangleq \mathop{\mathbb{E}}_{\tau \sim \mathcal{T}^T(\pi_1, \pi_2; G)} \left[ \frac{1}{T} \sum_{t \in [T]} U_i(\mathbf{x}^t, \mathbf{y}^t; G) \right].$$

We further define $\bar{U}_i^T(\pi_1, \pi_2; \mathcal{D}) \triangleq \mathbb{E}_{G \sim \mathcal{D}} \bar{U}_i^T(\pi_1, \pi_2; G)$ as the expected average utility under $\mathcal{D}$.

**Equilibrium in the Meta-Game.** We model the rationality of long-term players by treating the repeated Bayesian game $\mathcal{D}$ as a meta-game, where each player $P_i$'s action is an algorithm $\pi_i$, and the utilities of each pair of action $(\pi_1, \pi_2)$ are given by $\bar{U}_i(\pi_1, \pi_2; \mathcal{D})$. Our analysis focuses on the *pure Nash equilibria* (PNE) of this meta game applied to the asymptotic regime $T \to \infty$.

**Definition 2.1** (PNE of the Meta-Game). *We say that a pair of algorithms $(\pi_1, \pi_2)$ form a pure Nash equilibrium (PNE) in the meta-game if for all $i \in \{1, 2\}$ and all other algorithms $\pi_i'$,*

$$\limsup_{T \to \infty} \left( \bar{U}_i^T(\pi_i', \pi_{-i}; \mathcal{D}) - \bar{U}_i^T(\pi_i, \pi_{-i}; \mathcal{D}) \right) \leq 0,$$

*where $\pi_{-i}$ denotes the algorithm of $P_i$'s opponent.*

Finally, we define no-regret and no-swap regret algorithms below.

**Definition 2.2** (No-(Swap) Regret Algorithms). *An algorithm $\pi_1$ of $P_1$ is called* no-regret *if for all adversarial sequences $\mathbf{y}^{1:T} \in \Delta(\mathcal{A}_2)^T$, the strategies $\mathbf{x}^{1:T}$ output by $\pi_1$ satisfies*

$$\mathbb{E}[regret_1^T] \triangleq \mathbb{E}\left[ \max_{x^\star \in \mathcal{A}_1} \sum_{t \in [T]} U_1(x^\star, \mathbf{y}^t) - U_1(\mathbf{x}^t, \mathbf{y}^t) \right] \in o(T).$$

*Furthermore, $\pi_1$ is called* no swap-regret *if*

$$\mathbb{E}[swap\text{-}regret_1^T] \triangleq \mathbb{E}\left[ \max_{f : \mathcal{A}_1 \to \mathcal{A}_1} \sum_{t \in [T]} U_1(f(\mathbf{x}^t), \mathbf{y}^t) - U_1(\mathbf{x}^t, \mathbf{y}^t) \right] \in o(T),$$

*where $f(\mathbf{x}) \in \Delta(\mathcal{A}_1)$ denotes the mixed strategy induced by $f(x)$ as $x \sim \mathbf{x}$. We define no-(swap) regret algorithms for $P_2$ symmetrically.*

We remark that there exist no-regret and no-swap regret algorithms under both the full-information feedback and bandit feedback setting.

---

[2]Our results also hold in an alternative setting where both players can only play pure strategies at each round, i.e., $x^t \sim \mathbf{x}^t$ and $y^t \sim \mathbf{y}^t$. For the feedback $H_i^r$, the realized utilities and the observations about opponent's strategy should both be defined in terms of $(x^t, y^t)$ instead of $(\mathbf{x}^t, \mathbf{y}^t)$. As a result, $\tau(\pi_1, \pi_2; G, s_1, s_2)$ becomes a distribution over *pure-strategy* trajectories $(x^t, y^t)_{t \in [T]}$ instead of mixed strategies $(\mathbf{x}^t, \mathbf{y}^t)_{t \in [T]}$.

# 3 Interactions between fully-informed $P_1$ and partially-informed $P_2$

In this section, we analyze the setting with a fully-informed $P_1$ — i.e., $P_1$ knows the game $G$ being played — and a partially informed $P_2$. In other words, algorithms $\pi_1$ and $\pi_2$ can each take an observable signal as input, where the signals received by $P_1$ and $P_2$ are independently drawn from signal distributions $\varphi_{p_1}(\cdot|G)$ and $\varphi_{p_2}(\cdot|G)$ with precision $p_1 = 1$ and $p_2 < 1$, respectively. Recall that each $P_i$ sees her realized signal $s_i$ and knows the precision levels $p_1, p_2$ of both players' signals.

Our main takeaway is that there is a separation in the benchmarks for achievable cumulative utilities between $P_1$ and $P_2$ in this setting, when $P_1$ and $P_2$ employ algorithms that form a PNE of the meta-game. This is not surprising in the one-shot setting. But in the repeated setting, even with infinite rounds for $P_2$ to learn the game from feedback gained throughout the interaction, we show that there is still a separation in achievable benchmarks.

This separation could be due to two factors: 1) $P_2$'s inability to learn the game based on repeated interactions, and 2) $P_2$'s failure to achieve the benchmark utility despite successful learning. We discuss this in more detail in Section 3.2 and Appendix A. We show that if $P_2$ was able to learn the game based on external signals, then $P_2$ would be able to achieve the benchmark. This highlights a fundamental difference between learning based on interactions with the other player and learning independently without relying on the other player. In the latter scenario, a utility benchmark is always achievable, whereas in the former, it is sometimes unattainable.

The benchmark that we will show separates $P_1$ from $P_2$ is the average Stackelberg value with the player of interest as leader. Recall that $StackVal_i(\mathcal{D}) = \mathbb{E}_{G\sim\mathcal{D}}[StackVal_i(G)]$ for each $P_i$. We will demonstrate the separation by showing that $P_1$ is always able to achieve this benchmark through a PNE of the meta-game, for all $\mathcal{D}$, but there exists some distribution $\mathcal{D}$ in which no PNE of the meta-game yields $P_2$ her counterpart benchmark.

We will first state the theorems and provide proof sketches later. Our first theorem (Theorem 3.1) asserts that $P_1$ *can* achieve the benchmark $StackVal_1(\mathcal{D})$ by explicitly constructing a PNE pair of algorithms $(\pi_1, \pi_2)$ that grants $P_1$ this utility in the asymptotic regime. In the proof of this theorem, we provide the rate of convergence to this utility (Remark B.1).

**Theorem 3.1** (Benchmark achievable by $P_1$ for all $\mathcal{D}$). *For every game family $\mathcal{G}$ and every distribution $\mathcal{D} \in \Delta(\mathcal{G})$ supported on it, there exists an algorithm pair $(\pi_1, \pi_2)$ such that $(\pi_1, \pi_2)$ is a PNE of the meta-game, and $\forall G \in \mathcal{G}, \bar{U}_1^T(\pi_1, \pi_2; G) \geq StackVal_1(G) - o_T(1)$.*

*That is, for every realized game $G$, the expected average utility of $P_1$ over $T$ rounds tends to $StackVal_1(G)$ as $T \to \infty$. The expectation is over the trajectories — sequence of player strategies, and resulting utilities induced by the algorithms $\pi_1, \pi_2$ and $G$.*

The next theorem completes the separation argument by constructing a specific game distribution where no PNE of the meta-game allows $P_2$ to asymptotically achieve the benchmark $StackVal_2(\mathcal{D})$.

**Theorem 3.2** (Benchmark unachievable by $P_2$ for some $\mathcal{D}$). *For all thresholds $p^\star \in [0, 1)$, there exists a game family $\mathcal{G}$ and a distribution $\mathcal{D} \in \Delta(\mathcal{G})$, s.t., $\forall p_2 \leq p^\star$, all PNE $(\pi_1, \pi_2)$ of the meta-game where $P_2$'s signal is of precision $p_2$ must suffer $\mathbb{E}_{G\sim\mathcal{D}} \bar{U}_2^T(\pi_1, \pi_2; G) \leq StackVal_2(\mathcal{D}) - \Omega_T(1)$.*

*This implies that there is a game $G \in \mathcal{G}$ such that when $G$ is realized, $P_2$'s expected average utility over $T$ rounds remains strictly bounded below $StackVal_2(G)$ even as $T \to \infty$.*

Theorems 3.1 and 3.2 show that there is a separation in achievable benchmark whenever the less-informed player is at any informational disadvantage, however small, compared to the fully-informed player. $P_2$'s signal could be arbitrarily close to being fully informative (i.e., $p_2$ is arbitrarily close to 1), but there is still a barrier between what $P_2$ can achieve compared to $P_1$, when $P_1$ has full knowledge.

## 3.1 Proof sketches of main theorems

Now we present proof sketches for the two theorems, defering the full proofs to the appendices.

*Proof sketch of Theorem 3.1.* Our proof puts together results from previous work [17, 28]. We present the proofs of these results for completion. In this proof sketch, we will prove the theorem when every game $G$ in the support of $\mathcal{D}$ is such that $P_2$ has a unique best-response $y(\mathbf{x}^\star; G)$ to $P_1$'s optimal Stackelberg strategy $\mathbf{x}^\star(G)$. The full proof is in Appendix B.1.

Let $\pi_1$ be the algorithm that plays $P_1$'s optimal Stackelberg strategy of the realized game $G$ (i.e., $\mathbf{x}^\star(G)$) at every round. Since $P_1$ has access to a signal that fully reveals the realized game $G$, $P_1$ can compute $\mathbf{x}^\star(G)$ and employ this strategy. Let $\pi_2$ be a no-swap-regret algorithm in the bandit-feedback setting (the algorithm is only based on the utilities received in each round). Such algorithms exist [7, 16, 34] and are deployable by $P_2$ without any knowledge of the game played or $P_1$'s strategies. Note that $\pi_2$ does not use $P_2$'s signal $s_2$. Therefore our analysis holds for all levels of precision of $s_2$.

First, let us analyze the expected utility of $P_2$ due to $(\pi_1, \pi_2)$. The generated trajectories when the game $G$ is realized are of the form $(\mathbf{x}^\star(G), \mathbf{y}^t)_{t=1}^\infty$. Since $\pi_2$ is a no-swap-regret algorithm, the regret of this trajectory up to round $T$ is sub-linear in $T$ $(o(T))$.

Since we assumed that $\text{BR}_2(\mathbf{x}^\star; G)$ is unique, any round where $P_2$ is not employing this unique best-response $(y(\mathbf{x}^\star; G))$ causes $P_2$ to incur regret. The no-swap-regret property for $P_2$ essentially means that $P_2$'s strategies in the trajectory $(\mathbf{y}^t)_{t=1}^\infty$ become close to $y(\mathbf{x}^\star; G)$. And as a result, $P_1$'s utility per round gets close to $U_1(\mathbf{x}^\star(G), y(\mathbf{x}^\star; G); G)$ which is $\text{StackVal}_1(G)$.

More formally, $P_1$'s cumulative utility over $T$ rounds satisfies $\sum_{t \in [T]} U_1(\mathbf{x}^\star(G), \mathbf{y}^t) \geq \text{StackVal}_1(G) \cdot T - c_1 \sum_{t \in [T]} \|\mathbf{y}^t - \mathbf{y}(\mathbf{x}^\star; G)\|_1$, where $\mathbf{y}(\mathbf{x}^\star; G)$ is the one-hot vector encoding of $y(\mathbf{x}^\star; G)$ and $c_1 = \max_{y \in \mathcal{A}_2 \setminus \{y(\mathbf{x}^\star; G)\}} U_1(\mathbf{x}^\star(G), y; G)$. We bound term $\sum_{t \in [T]} \|\mathbf{y}^t - \mathbf{y}(\mathbf{x}^\star; G)\|_1$ using the no-swap-regret property. $P_2$'s swap regret is at least $\sum_{t \in [T]} c_2 \|\mathbf{y}^t - \mathbf{y}(\mathbf{x}^\star; G)\|_1$, where $c_2 = U_2(\mathbf{x}^\star(G), y(\mathbf{x}^\star; G)) - \max_{y \in \mathcal{A}_2 \setminus \{\mathbf{y}(\mathbf{x}^\star; G)\}} U_2(\mathbf{x}^\star(G), y)$ is the minimum difference of $P_2$'s utility between playing the best response action $\mathbf{y}(\mathbf{x}^\star; G)$ and any other action in $\mathcal{A}_2$. Sub-linear swap regret therefore implies that $\mathbb{E}\left[\sum_{t=1}^{T} \|\mathbf{y}^t - \mathbf{y}(x^\star; G)\|_1\right] \in o(T)$ and thus $\mathbb{E}\left[\sum_{t=1}^{T} U_1(\mathbf{x}^\star(G), \mathbf{y}^t)\right] \geq \text{StackVal}_1(G) \cdot T - o(T)$, i.e., $P_1$'s expected average utility in $T$ rounds is at least $\text{StackVal}_1(G) - o_T(1)$.

We have shown that the pair $(\pi_1, \pi_2)$ achieves $P_1$'s benchmark utility. We now show that it is a PNE of the meta-game. Fixing $\pi_1$, the maximum utility $P_2$ can get is the utility achieved by playing $y(\mathbf{x}^\star; G), \forall t$. $P_2$ does not necessarily know $G$ to play $y(\mathbf{x}^\star; G)$ for all $t \in [T]$, but we have shown that due to $\pi_2$ being a no-swap-regret algorithm, $P_2$ ends up playing strategies close to $y(\mathbf{x}^\star; G)$ asymptotically. The difference between $P_2$'s cumulative utility between playing $\pi_2$ against $\pi_1$, versus playing per-round best response against $\pi_1$ is at most $O(\sum_{t \in [T]} \mathbb{E} \|\mathbf{y}^t - \mathbf{y}(\mathbf{x}^\star; G)\|_1)$ which is $o(T)$ by the no-swap-regret property. So $P_2$ has vanishing incentive to deviate from $\pi_2$ in the meta-game.

Next fixing $\pi_2$ to be a no-swap-regret algorithm, previous work [17, 28] caps $P_1$'s achievable utility through any algorithm $\pi_1'$ (Deng et al. [17, Theorem 6]). These results show that for every $\pi_1'$, $P_1$'s expected average utility induced by $(\pi_1', \pi_2)$ in $T$ rounds is at most $\text{StackVal}_1(G) + o_T(1)$. Since we have shown that $(\pi_1, \pi_2)$ yields at least $\text{StackVal}_1(G) - o_T(1)$ for $P_1$, there is vanishing incentive for $P_1$ to deviate.

In Appendix B.1, we extend this proof to the scenario with potential ties in $P_2$'s best response, but under the assumption that $P_2$ has no weakly dominated action. Using regret rates of standard swap-regret algorithms, we also provide the rate of convergence to the Stackelberg benchmark. □

*Proof sketch of Theorem 3.2.* To prove this theorem, we construct a family of two games $G_1$ and $G_2$ (shown in Figure 1) and let the prior distribution $\mathcal{D}$ to be uniform over $G_1$ and $G_2$. Note that the maximum value of game parameters depends inversely on $\gamma \triangleq \frac{1-p^\star}{1+p^\star}$, where $p^\star$ is the maximum precision of the signal received by $P_2$. In this construction, the utility functions in both games are identical for $P_2$ but different for $P_1$. This implies that $P_2$ cannot gain any additional knowledge about which game is realized from looking at her own utility function.

We first illustrate the high-level idea by considering a hypothetical situation where the trajectory always converges to the Stackelberg equilibrium led by $P_2$ for all $G$. In other words, the trajectory converges to $(x(\mathbf{y}^\star; G_1), \mathbf{y}^\star(G_1))$ when $G_1$ is realized and $(x(\mathbf{y}^\star; G_2), \mathbf{y}^\star(G_2))$ when $G_2$ is realized. It is not hard to check that the Stackelberg equilibria turns out to be supported on different pure-strategy pairs: $(A, C)$ in $G_1$ and $(B, D)$ in $G_2$ (shaded cells in Figure 1). Because the Stackelberg strategies differ for $G_1$ and $G_2$, to converge to the correct equilibrium, $P_2$ must have gained full information about which game $G$ is being played through repeated interactions with $P_1$. However, from $P_1$'s perspective, the strategy pair $(A, C)$—the Stackelberg equilibrium led by $P_2$ in $G_1$—is

|   | C | | D | |
|---|---|---|---|---|
| A | $16/\gamma,$ | $1$ | $16/\gamma,$ | $-32/\gamma$ |
| B | $2,$ | $0$ | $0,$ | $2$ |

Game Matrix $G_1$

|   | C | | D | |
|---|---|---|---|---|
| A | $1,$ | $1$ | $0,$ | $-32/\gamma$ |
| B | $0.9,$ | $0$ | $0.1,$ | $2$ |

Game Matrix $G_2$

Figure 1: game matrices $G_1$ and $G_2$. $P_1$ is the row player and $P_2$ is the column player. The values in each cell are ($P_1$'s utility, $P_2$'s utility). Shaded cells represent the action profiles supported in the Stackelberg equilibria led by the column player. The parameter $\gamma$ is defined as $\frac{1-p^\star}{1+p^\star} \in (0,1]$, where $p^\star$ is the precision threshold of $p_2$.

more favorable than the other equilibrium $(B, D)$ in both $G_1$ and $G_2$. Therefore, instead of disclosing information about which $G$ is realized, it would be more beneficial for $P_1$ to conceal this information and always behave as if $G$ were $G_1$. Therefore, any pair of algorithms that give rise to this hypothetical situation cannot be an equilibrium in the space of algorithms.

Our actual proof applies similar ideas to establish a stronger claim: not only is it impossible for $P_2$ to have the trajectory always converge to their Stackelberg equilibrium, but they cannot recover an average utility of $\text{StackVal}_2(\mathcal{D})$ through *any* repeated interactions with $P_1$ that are specified by PNE algorithm pairs. To argue this, we will use the notion of *correlated strategy profiles (CSP)* [3] as a succinct way of analyzing the expected utility of each player. For a distribution $\mathcal{T}^T$ over trajectories of length $T$, the CSP induced by $\mathcal{T}^T$, denoted as $\text{CSP}_{\mathcal{T}^T}$, is a correlated distribution in $\Delta(\mathcal{A}_1 \times \mathcal{A}_2)$ which is taken as the empirical average of the mixed-strategy profiles in each time step, i.e., $\text{CSP}_{\mathcal{T}^T} \triangleq \mathbb{E}_{(\mathbf{x}_t, \mathbf{y}_t)_{t \in [T]} \sim \mathcal{T}^T}[(1/T) \sum_{t \in [T]} \mathbf{x}_t \otimes \mathbf{y}_t]$. Since CSPs serve as a sufficient statistics of both players' expected utility (which is a direct consequence of the linearity of utilities), working with them significantly reduces the dimension of the problem.

**Special case: full information asymmetry.** We start with the full information asymmetry setting, i.e., $p_1 = 1$ and $p_2 = 0$. For the sake of contradiction, assume that a pair of equilibrium algorithms $(\pi_1, \pi_2)$ can let $P_2$ achieve the benchmark $\text{StackVal}_2(\mathcal{D}) = 3/2$. With the CSPs introduced above, we can rewrite $P_2$'s average expected utility as $\frac{1}{2}\mathbb{E}_{(x,y) \sim \text{CSP}_1} U_2(x, y; G_1) + \frac{1}{2}\mathbb{E}_{(x,y) \sim \text{CSP}_2} U_2(x, y; G_2)$, where we have used $\text{CSP}_1$ and $\text{CSP}_2$ to denote the CSPs induced by the distribution over trajectories generated by $\mathcal{T}^T(\pi_1, \pi_2; G_1)$ and $\mathcal{T}^T(\pi_1, \pi_2; G_2)$, respectively.

Similar to the hypothetical situation sketched above, we want to argue that there is incentive for $P_1$ to deviate to an algorithm $\pi_1'$ that always behaves according to $\pi_1(G_1)$ even when the actual game is $G_2$. In other words, we aim to show that $P_1$'s expected utility in $G_2$ strictly increases after replacing the induced CSP from $\text{CSP}_2$ to $\text{CSP}_1$, i.e., $\mathbb{E}_{\tau \sim \text{CSP}_1} U_1(\tau; G_2) > \mathbb{E}_{\tau \sim \text{CSP}_2} U_1(\tau; G_2)$. Note that for $P_1$'s utility in $G_2$, cells involving action $C$ all have utility close to 1, whereas those involving action $D$ all have utility close to 0. Therefore, it suffices to show that cells involving action $D$ take up a significant probability mass in $\text{CSP}_2$ but very little in $\text{CSP}_1$. We break these into the following three claims and use the equilibrium condition to establish them in Appendix B.3.

- **Claim 1.** $\text{CSP}_1(B, D)$ is very small, otherwise $P_1$ would deviate to always playing action $A$.
- **Claim 2.** $\text{CSP}_1(A, D)$ is very small, otherwise $P_2$ would deviate to always playing action $C$.
- **Claim 3.** $\text{CSP}_2(B, D)$ is very large, otherwise $P_2$ cannot achieve benchmark $\text{StackVal}_2(\mathcal{D})$.

**Towards partial information asymmetry.** In the remainder of this sketch, we discuss the extension of the above approach to the partial asymmetry setting where $p_1 = 1$ and $0 \leq p_2 \leq p^\star < 1$. The fact that $P_2$'s signal is partially informative introduces extra challenge to our analysis, since $P_2$'s belief about the true game depends not only on $P_1$'s behavior during the interaction, but also on the information carried by the external signal $s_2$. As a result, if $P_1$ deviates to acting according to $G_1$ when the actual game is $G_2$, it does not trigger the expected CSP when the realized game is $G_1$, but instead causes a "distorted" posterior since the distribution of $s_2 \sim \varphi_{p_2}(\cdot|G_2)$ does not change.

To illustrate this, consider the four different CSPs introduced by all combinations of the realized signals received by both players. For $(i, j) \in \{1, 2\}^2$, let $\text{CSP}_{ij}$ to denote the CSP induced by $\mathcal{T}^T(\pi_1, \pi_2; s_1, s_2, G = s_1)$ when $s_1 = G_i$ and $s_2 = G_j$ (we have set $G = s_1$ because $s_1$ perfectly reveals $G$). When the realized game is $G_2$, $P_1$'s expected utility before deviation is given by

$$\bar{U}_1(\pi_1, \pi_2; G_2) = \frac{1 - p_2}{2} \mathbb{E}_{\tau \sim \text{CSP}_{21}} U_2(\tau; G_2) + \frac{1 + p_2}{2} \mathbb{E}_{\tau \sim \text{CSP}_{22}} U_2(\tau; G_2),$$

because the probability of $P_2$ seeing signals $s_2 = G_1$ and $s_2 = G_2$ are $\frac{1-p_2}{2}$ and $\frac{1+p_2}{2}$, respectively. So, as for the expected utility after deviation, the coefficients $\frac{1-p_2}{2}$ and $\frac{1+p_2}{2}$ remain the same, but the first distribution $\mathsf{CSP}_{21}$ becomes $\mathsf{CSP}_{22}$ and the second distribution changes from $\mathsf{CSP}_{22}$ to $\mathsf{CSP}_{12}$:

$$\bar{U}_1(\pi'_1, \pi_2; G_2) = \frac{1-p_2}{2} \mathop{\mathbb{E}}_{\tau \sim \mathsf{CSP}_{11}} U_2(\tau; G_2) + \frac{1+p_2}{2} \mathop{\mathbb{E}}_{\tau \sim \mathsf{CSP}_{12}} U_2(\tau; G_2),$$

However, if the true game were $G_1$, then $\mathsf{CSP}_{11}$ and $\mathsf{CSP}_{12}$ would be realized with swapped probability $\frac{1+p_2}{2}$ and $\frac{1-p_2}{2}$, not the ones appeared in $\bar{U}(\pi'_1, \pi_2; G_2)$! Hence, even if we can guarantee that action pairs $(A, D)$ and $(B, D)$ occur very infrequently when the true game is $G = G_1$, they may only occur under $\mathsf{CSP}_{12}$, whose frequency gets amplified by $\frac{1+p_2}{1-p_2}$ times when factoring into the utility after deviation $\bar{U}(\pi'_1, \pi_2; G_2)$. Therefore, establishing the benefit of deviation requires a much smaller probability of $(A, D)$ and $(B, D)$ under the CSPs induced by $G = G_1$. This is why we need the game parameters to inversely depend on $\gamma = \frac{1-p^\star}{1+p^\star}$, where $p^\star$ is an upper bound on $p_2$. $\qquad\square$

## 3.2 Difference in learning through repeated interactions and learning independently

In this section, we provide an informal discussion on the reason behind $P_2$'s failure to recover their Stackelberg value benchmark through repeated interactions, with a more formal treatment deferred to Appendix A. We argue that the failure is not due to the PNE of meta-game always preventing $P_2$ from "learning" the game, but rather because $P_2$ cannot apply her learned knowledge to recover her Stackelberg value in any equilibrium.

At each round, $P_2$ can form a posterior belief about the realized game $G$ based on her observed feedback from the historical interactions and the initial signal $s_2$ received. We say that $P_2$ *successfully learns* $G$ if her posterior belief converges to the point distribution on $G$ (formally in Definition A.2). Interestingly, using the same pair of PNE algorithms designed to show that $P_1$ can achieve their Stackelberg value, we can show that $P_2$ *is indeed* able to successfully learn $G$ through strategic interactions. This is because $P_2$'s strategy converges to the best response $y(\mathbf{x}^\star; G)$, which perfectly reveals $G$ for some game families $\mathcal{G}$. Thus, successful learning of $G$ through repeated interactions *can* happen in a PNE of a meta-game where $\mathrm{StackVal}_2(\mathcal{D})$ cannot be achieved (Observation A.4).

The problem preventing $P_2$ from achieving $\mathrm{StackVal}_2(\mathcal{D})$ is not an insufficient rate or accuracy of learning, but rather the fact that learning and acting on this learned knowledge are intertwined. In fact, if $P_2$'s learning was independent of the repeated interaction, i.e., when $P_2$ has access to external signals that become more accurate over time, she *can* achieve $\mathrm{StackVal}_2(\mathcal{D})$ (Proposition A.5).

## 4 Interactions between two partially-informed players

In this section, we consider the setting where neither player is fully informed. That is, the precision of both player's signals $(p_1, p_2)$ are less than one. Even though there may be information asymmetry in the form of different precision levels of player signals, we show that there is no longer a clear separation between players through the average Stackelberg value benchmark.

At a high level, what distinguishes this setting from the previous setting (hence resulting in the lack of separation), is that the identity of the more-informed player can shift throughout the course of the repeated interaction. Due to the structure of $\mathcal{D}$, more information about the realized game may be released to one player compared to the other. In contrast, when the more informed player starts with perfectly knowing the realized game, there is no possibility of her becoming less informed since there is no information beyond what she already knows.

**Example 4.1.** *Consider $\mathcal{D}$ to be the uniform distribution over the two game matrices defined in the figure below.*

|   | $C$ | $D$ |
|---|---|---|
| $A$ | 1,  1 | $-1$,  5 |
| $B$ | 0,  2 | 2,  5 |

Game Matrix $G_1$

|   | $C$ | $D$ |
|---|---|---|
| $A$ | 1,  3 | $-1$,  0 |
| $B$ | 0,  7 | 2,  8 |

Game Matrix $G_2$

Figure 2: Example game matrices $\mathcal{G} = \{G_1, G_2\}$ revealing more information to $P_2$ compared to $P_1$. Here, $P_1$ is the row player and $P_2$ is the column player.

*Note that if* $P_2$ *chooses the pure strategy* $C$, *then for any strategy of* $P_1$, $P_2$'s *utility lies in the range* $[1, 2]$ *if* $G_1$ *is realized and in the range* $[3, 7]$ *if* $G_2$ *is realized. Since both ranges are non-intersecting,* $P_2$ *can deduce* $G$ *exactly after a single round by choosing the pure strategy* $B$ *in the first round.*

*However, since the utilities of* $P_1$ *for all action profiles are the same in both* $G_1, G_2$, $P_1$ *gains no additional information about the realized game. So even if* $P_1$ *started off with a more informative signal, after a single round,* $P_2$ *becomes more informed and in fact perfectly informed.*

To show that the average Stackelberg value benchmark does not separate the more- from the less-informed player, we will show that neither player can achieve the average Stackelberg value in all instances. Put another way, for every possible pair of player signals' precision, there is an instance such that this player cannot achieve the benchmark value at equilibrium.

**Proposition 4.2.** *For every player signal precision values* $p_1, p_2 \in [0, 1)$, *for each* $i \in \{1, 2\}$, *there exists* $\mathcal{D}$ *such that for every PNE* $(\pi_1, \pi_2)$ *of the meta-game,* $\bar{U}_i^T(\pi_1, \pi_2; \mathcal{D}) \leq StackVal_i(\mathcal{D}) - \Omega_T(1)$.

*Proof.* The proof of this proposition reduces to the proof of Theorem 3.2. This is due to the following game-revealing property (similar to Example 4.1) of the construction $\mathcal{D}$ used in the proof of Theorem 3.2 described by Figure 1. Since the range of the set of attainable utilities in $G_1, G_2$ when $P_1$ chooses action $A$, has no intersection for $P_1$, regardless of $P_2$'s strategy, $P_1$ can deduce the game exactly after a single round while $P_2$ gains no additional information after a single round.

After the first round, we are in the regime of a fully informed $P_1$ and a partially informed $P_2$ since $p_2 < 1$. Theorem 3.2 already shows that in this regime, no equilibrium provides $P_2$ her average Stackelberg value benchmark. Using a distribution $\mathcal{D}'$ that is the same as $\mathcal{D}$ but with player utilities flipped proves the proposition for $P_2$. $\square$

# 5 Discussion

In this paper, we study the effects of information asymmetry (codified in terms of signals about the game played) on the achievable benchmarks of two non-myopic players interacting repeatedly over $T$ rounds. First, we showed that when $P_1$ is fully informed (i.e., knows $G$) while $P_2$ is not, then there is a separation between the more and the less informed player by way of each player's achievable benchmarks. Next, we showed that when neither player is fully informed (i.e., both $p_1, p_2 < 1$) then, there is no longer a clear separation between players in terms of benchmarks.

There are several avenues for future research stemming from our work.

**Characterizations of algorithms that can be supported in an equilibrium.** We should gain a better understanding of what algorithms from natural classes can be in equilibrium. A useful step would be to characterize the necessary and/or sufficient conditions for an algorithm to be supported in the PNE of the meta-game. Our work and previous work provide sufficient conditions such as no-swap-regret algorithms and best-responding per round: an algorithm satisfying either condition can be supported in a PNE of every meta-game if at least one player is fully informed. Previous work also implies that no-regret is not sufficient for an algorithm to be part of a meta-game PNE (see Appendix C for more details). Finally, in the case where neither player is fully informed, it would be very useful to characterize the structure of the meta-game that causes a shift wrt the information advantage.

**Other models of how signals are generated.** Alleviating some of our modeling assumptions, one could ask how the results would change if nature was not assumed to be truthful with respect to the signal reporting but it may strategically modify the signals to achieve its own goals, such as maximizing social welfare. Taking this aspect into account, we can consider an information design setting where the nature designs a signaling scheme that shapes both agents' beliefs about the state and therefore their algorithms of choice. In addition, we have assumed that nature provides signals cost-free. This is a required and natural first step, but an interesting direction would be to understand what happens when the signals are costly and their accuracy is positively correlated with their cost.

**Computational aspects of meta-game equilibrium.** Moreover, it would be very interesting to see how the results about the effects of information asymmetry generalize in the case where the players are computationally bounded; note that our current setup provides information-theoretic results, but it could be computationally hard for players to communicate their algorithms to each other, or even verify that two algorithms are at equilibrium.

**Acknowledgements**

This work was supported in part by the National Science Foundation under grant CCF-2145898, by the Office of Naval Research under grant N00014-24-1-2159, a C3.AI Digital Transformation Institute grant, an Alfred P. Sloan fellowship, a Schmidt Science AI2050 fellowship, and an Amazon Research Award.

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

# A Difference in learning through repeated interactions and learning independently

In this part, we will use Theorem 3.2 to show that there is a difference between learning through repeated interactions and learning independently based on external signals. We will argue that learning independently is more powerful as it allows achieving the average Stackelberg benchmark, whereas learning based on repeated interactions does not always allow this.

Our approach to doing this is to introduce two models of learning: one based on histories generated by the repeated interaction and the other based on external signals. Fixing the same success criterion for both models (to be defined soon), we will show that

1. Successful learning by $P_2$ based on repeated interactions is possible at a PNE of the meta-game, but still does not yield $P_2$ her average Stackelberg value (Observation A.4).

2. Successful learning by $P_2$ based on external signals makes $P_2$'s average Stackelberg value achievable at a PNE of the meta-game, for all $\mathcal{D}$ (Proposition A.5).

We first define both learning models and the success criterion of learning. Later in this section, we state results demonstrating the separation between the two learning models.

Successfully learning a realized game $G$ entails forming beliefs over the realized game such that the beliefs asymptotically concentrate on the true realized game $G$.

**Definition A.1** (Successful learning criterion). *Given a realized game $G$ from a game family $\mathcal{G}$, a belief sequence $(\hat{\beta}^t)_{t=1}^{\infty}$, where $\hat{\beta}^t \in \Delta(\mathcal{G})$ is a belief (distribution) over $\mathcal{G}$ for each $t \in \mathbb{N}$, successfully learns $G$ if $\Pr_{\tilde{G}^T \sim \hat{\beta}^T}[\tilde{G}^T \neq G] \in o_T(1)$. That is, the beliefs asymptotically fully concentrate on the realized game $G$.*

Different models of learning involve different constraints on or power afforded to how beliefs of the realized game are generated. The first model of interest is learning based on repeated interactions. Here, the belief at a round $t$ is constrained to be formed based on the initial signal and the history at round $t$ generated during the repeated interaction.

**Definition A.2** (Successful learning based on repeated interactions). *Given a family of games $\mathcal{G}$ and a player $P_i$ ($i \in \{1, 2\}$), a history-based belief function for $P_i$ is a mapping from the player's signal value and a history of repeated interactions to a belief distribution supported on $\mathcal{G}$. Formally, we denote the belief function with $h : (s_i; H_i^{1:t-1}) \mapsto \Delta(\mathcal{G})$.*

*Given a distribution $\mathcal{D} \in \Delta(\mathcal{G})$, we say a history-based belief function $h$, as defined above, successfully learns based on interactions through the algorithm pair $(\pi_1, \pi_2)$ if for every realized game $G \in \mathcal{G}$ and every realized signal $s_1, s_2$, the induced beliefs $\left(\hat{\beta}_i^t\right)_{t=1}^{\infty}$ where each belief $\hat{\beta}_i^t = h(s_i, H_i^{1:t-1})$ is induced by histories $H_i^\tau$ generated from the distribution $\mathcal{T}^T(\pi_1, \pi_2; G, s_1, s_2)$, successfully learns the realized game $G$ (according to Definition A.1) with probability 1.*

The second form of learning occurs without dependence on the other player's actions and instead based on externally provided signals.

**Definition A.3** (Successful independent learning). *Given a distribution of games $\mathcal{D}$, we say that $P_i$ can independently learn successfully if for every realized game $G \sim \mathcal{D}$, there is a sequence of signals $(q^t)_{t=1}^{\infty}$ with $q^t \in \mathcal{G}$, that are generated by a sequence of signaling distributions $(Q^t)_{t=1}^{\infty}$ with $Q^t \in \Delta(\mathcal{G})$, where the sequence $(Q^t)_{t=1}^{\infty}$ successfully learns the realized game $G$ (according to Definition A.1).*

*The signal $q^t$ is provided to $P_i$ at round $t$. So $P_i$'s algorithm $\pi_i$ maps the initial signal $s_i$, history at each round $t$ ($H_i^{1:t-1}$), and $q^t$ to a strategy taken at round $t$.*

Revisiting Theorem 3.2, which states that the average Stackelberg value is not achievable by $P_2$ for some $\mathcal{D}$, we can question whether some property of $\mathcal{D}$ and the equilibria of the meta-game prevents $P_2$ from successfully learning or if $P_2$ can successfully learn but cannot use this learning to achieve her Stackelberg value. We assert that it is the latter.

**Observation A.4.** *There is a game distribution $\mathcal{D}$, such that there exists a PNE $(\pi_1, \pi_2)$ of the meta-game that allows $P_2$ to successfully learn based on repeated interactions (in the sense of*

*Definition A.2), but no equilibrium allows* $P_2$ *to achieve her average Stackelberg value benchmark* $StackVal_2(\mathcal{D})$.

|   | D | | E | | F | |
|---|---|---|---|---|---|---|
| A | $16/\gamma,$ | $1-\epsilon$ | $16/\gamma,$ | $-32/\gamma$ | $16/\gamma,$ | $1+\epsilon$ |
| B | $2,$ | $0$ | $0,$ | $2$ | $2,$ | $0$ |
| C | $(16-\epsilon)/\gamma,$ | $1+\epsilon$ | $16/\gamma,$ | $-32/\gamma$ | $(16-\epsilon)/\gamma,$ | $1-\epsilon$ |

Game Matrix $G_1'$

|   | D | | E | | F | |
|---|---|---|---|---|---|---|
| A | $1,$ | $1-\epsilon$ | $0,$ | $-32/\gamma$ | $1,$ | $1+\epsilon$ |
| B | $0.9,$ | $0$ | $0.1,$ | $2$ | $0.9,$ | $0$ |
| C | $1+\epsilon,$ | $1+\epsilon$ | $0,$ | $-32/\gamma$ | $1+\epsilon,$ | $1-\epsilon$ |

Game Matrix $G_2'$

Figure 3: game matrices $G_1'$ and $G_2'$. $P_1$ is the row player and $P_2$ is the column player. The values in each cell are ($P_1$'s utility, $P_2$'s utility). The parameter $\gamma$ is defined as $\frac{1-p^\star}{1+p^\star} \in (0, 1]$, where $p^\star$ is the precision threshold of $p_2$.

*Proof.* Consider the construction of $\mathcal{D}$ used in Theorem 3.2 (Figure 1). $\mathcal{D}$ is a distribution with equal probabilities over two game matrices $G_1, G_2$. Theorem 3.2 proves that the average Stackelberg benchmark is unattainable by $P_2$ in any equilibrium.

For this proof, we will consider another construction $\mathcal{D}'$ that is equal probability over game matrices $G_1', G_2'$ defined in Figure 3. $G_1'$ is essentially $G_1$ with an additional row and column, where the additional row is essentially a duplicate of the first row and the additional column is essentially a duplicate of the first column. Rather than being an exact duplicate, the new row/column is a small perturbation (given by parameter $\epsilon$) of the original row/column it duplicates. $G_2'$ is obtained from $G_2$ similarly.

Since the new construction is essentially duplicating rows/columns of the old construction, by the same argument as is Theorem 3.2, the average Stackelberg benchmark is also unattainable by $P_2$ in any equilibrium of the meta-game of the Bayesian game $\mathcal{D}'$.

Despite this, we can show there is an equilibrium enabling $P_2$ to successfully learn through repeated interactions. We construct the perturbations to ensure that the $P_1$-led equilibrium response has a different response for $P_2$ in $G_1'$ compared to $G_2'$. Our argument is that the meta-game PNE pair results in both players playing the $P_1$-led Stackelberg equilibrium of the realized game. Therefore, based on $P_2$'s responses generated by the trajectories of the meta-game PNE, $P_2$ can determine the realized game.

Consider the algorithm pair where $P_1$ plays her Stackelberg strategy of the realized game and $P_2$ plays a no-swap-regret algorithm. The proof of Theorem 3.1 showed that this pair forms an equilibrium and that $P_2$'s strategy eventually becomes the best-response of the realized game due to $P_2$'s no-swap-regret property. Note that the two games $G_1', G_2'$ in the support of $\mathcal{D}'$ have different $P_2$ responses in $P_1$'s Stackelberg equilibrium. So based on the strategies $P_2$ plays, she will be able to deduce the realized game. Consider the belief function $h$ that at round $t$ computes the average strategy employed thus far: $\bar{\mathbf{y}}_t = 1/(t-1)\sum_{r=1}^{t-1} \mathbf{y}_r$ and forms a belief concentrated on $G_1'$ if $\|\bar{\mathbf{y}}_t - \mathbf{y}(\mathbf{x}^\star; G_1')\| < \|\bar{\mathbf{y}}_t - \mathbf{y}(\mathbf{x}^\star; G_2')\|$ and forms a belief concentrated on $G_2'$ otherwise, where $\mathbf{y}(\mathbf{x}^\star; G_i')$ is the one-hot encoding vector of the best response $y(\mathbf{x}^\star; G)$ in game $G$. The induced sequence of beliefs concentrates on the true realized game and hence satisfies the accuracy criterion for learning the realized game. Therefore, there is an equilibrium of algorithms that allows $P_2$ to successfully learn.

$\square$

The above observation highlighted a limitation of successful learning based on repeated interactions. There could be two reasons driving this limitation. One possibility is that the accuracy criterion for successful learning is not strong enough to always enable achieving the average Stackelberg value benchmark. The other possibility is that the learning process relies on the actions of the other player in the repeated interaction.

We will now argue that the limitation arises from the second reason and not the first. If the accuracy criterion for successful learning is met through independent learning based on external signals instead of histories of the game dynamics, we will show that the average Stackelberg value benchmark becomes achievable.

The following proposition shows that if $P_2$ can successfully learn independently for every $\mathcal{D}$, then $P_2$ can achieve her average Stackelberg value $StackVal_2(\mathcal{D})$ for every $\mathcal{D}$ at a PNE of the meta-game. This extends Theorem 3.1, which stated that this is possible if $P_2$ was fully informed. Essentially, successful independent learning guarantees the same power to a player as being fully informed from the beginning.

**Proposition A.5.** *For every $\mathcal{D}$ for which $P_2$ can successfully learn independently (Definition A.3), there is an algorithm pair $(\pi_1, \pi_2)$ such that $(\pi_1, \pi_2)$ is a PNE of the meta-game and $\forall G \in \mathcal{G}$, $\bar{U}_2(\pi_1, \pi_2; G) \geq StackVal_2(G) - o_T(1)$.*

*Proof sketch.* The equilibrium algorithm pair we will show guarantees this utility to $P_2$ is the one where $P_2$ employs the Stackelberg response of the game denoted by the external signal $q^t$ at round $t$ and $P_1$ employs a no-swap-regret algorithm. $\qquad\square$

We defined successful independent learning as when the less informed player learns the realized game to arbitrarily high precision, based on external signals. If the external signals were not quite as powerful, the average Stackelberg benchmark can no longer be always achieved. In particular, if $P_2$ could only learn the realized game to a precision bounded away from 1, Theorem 3.2 shows that the average Stackelberg benchmark is not always achievable when $P_1$ is fully informed about the game.

# B  Missing Proofs

## B.1  Proof of Theorem 3.1

Theorem 3.1 is implied by Proposition A.5. Please see the proof of Proposition A.5 in Appendix B.2.

## B.2  Proof of Proposition A.5

We will construct an algorithm pair $(\pi_1, \pi_2)$ such that when $P_1$ is able to successfully learn through an independent sequence of signals $(q^t)_{t=1}^{\infty}$, $(\pi_1, \pi_2)$ is an equilibrium pair and guarantees $P_1$ an expected average utility of at least $StackVal_1(G) - o_T(1)$ in $T$ rounds, for every realized game $G$. The same proof also holds if $P_2$ is the player able to learn through external signals. Let $(\mathbf{x}^{\star}(G), y(\mathbf{x}^{*}; G))$ denote $P_1$'s Stackelberg strategy and $P_2$'s best response in the game $G$. Let $\mathbf{y}(\mathbf{x}^{*}; G)$ be the one-hot encoding vector of $y(\mathbf{x}^{*}; G)$.

We choose $\pi_2$ to be a no-swap-regret algorithm with swap-regret rate $O(T^a)$ for $a < 1$. We will first construct $\pi_1$ in the setting with a known, finite time horizon $T$ so that $P_1$ achieves average regret $StackVal_1(G) - o_T(1)$ in $T$ rounds when interacting with $P_2$ employing a no-swap-regret algorithm. This is builds on the proof of Theorem 4 by Deng et al. [17]. We will present the argument here for completeness. We will later show how to apply the doubling trick to extend this to the setting with infinite time horizon.

Under the assumption that the realized game has no weakly dominated actions, we will show how $P_1$ can choose a strategy $\mathbf{x}'$ so that $y(\mathbf{x}^{\star}; G)$ is $P_2$'s unique best response and $\mathbf{x}'$ is close to $\mathbf{x}^{\star}(G)$. Since $y(\mathbf{x}^{\star}; G)$ is not weakly dominated, there is no $\mathbf{y}' \in \Delta(\mathcal{A}_2 \setminus \{y(\mathbf{x}^{\star}; G)\})$ such that $U_2(\mathbf{x}, \mathbf{y}') \geq U_2(\mathbf{x}, y(\mathbf{x}^{\star}; G))$ for all $x \in \mathcal{A}_1$. By Farkas' lemma [20], there must exist a $\bar{\mathbf{x}} \in \Delta(\mathcal{A}_1)$ such that $U_2(\bar{\mathbf{x}}, y(\mathbf{x}^{\star}; G)) \geq U_2(\bar{\mathbf{x}}, y(\mathbf{x}^{\star}; G)) + c$ for a constant $c > 0$. This implies that if $P_1$ plays the strategy $\mathbf{x}'_{\delta} = (1 - \delta)\mathbf{x}^{\star} + \delta\bar{\mathbf{x}}$, then for all values of $\delta > 0$, $P_2$'s unique best response to $\mathbf{x}'_{\delta}$ is $y(\mathbf{x}^{\star}; G)$. To indicate dependence of this strategy on the game $G$, we will also denote it by $\mathbf{x}'_{\delta}(G)$. Choosing a small $\delta$ enables $P_1$ to play a strategy close to the Stackelberg strategy while ensuring that $P_2$'s unique best response is $y(\mathbf{x}^{\star}; G)$.

In the each round $t \in [T]$, $P_1$ employs $\mathbf{x}'_{\delta^T}(q^t)$, where $q^t$ is the signal at round $t$. We will later describe how to choose $\delta^T$. Let $(\mathbf{x}^t, \mathbf{y}^t)_{t=1}^{\infty}$ be the sequence of strategies generated through the interaction of the above algorithm of $P_1$ and $P_2$'s no-swap-regret algorithm. Given a $\delta^T$, let us compute $P_1$'s expected cumulative utility. Let $Z^t$ be a random variable indicating if the external signal at round $t$ is the realized game i.e., $Z^t = \mathbb{1}\{q^t = G\}$. In rounds with $Z^t = 1$, the immediate regret of $P_2$ at round $t$ is $c\|\mathbf{y}^t - \mathbf{y}(\mathbf{x}^{\star}; G)\|$. A lower bound on $P_2$'s regret based on regret accumulated

only in rounds with $Z^t = 1$ is $\sum_{t=1}^T \delta^T c \cdot Z^t \|\mathbf{y}^t - \mathbf{y}(\mathbf{x}^\star; G)\|$, where $c = U_2(\mathbf{x}', y(\mathbf{x}^\star; G)) - \max_{y \in \mathcal{A}_2 \setminus \{y(\mathbf{x}^\star; G)\}} U_2(\mathbf{x}', y)$.

$$\text{swap-regret} \geq \sum_{t=1}^T \delta^T c \cdot Z^t \|\mathbf{y}^t - \mathbf{y}(\mathbf{x}^\star; G)\|$$

$$\mathbb{E}[\text{swap-regret}] \geq \sum_{t=1}^T \delta^T c \cdot \mathbb{E}[Z^t] \cdot \mathbb{E}[\|\mathbf{y}^t - \mathbf{y}(\mathbf{x}^\star; G)\|]$$

$$\text{(Independence between } q^t \text{ and P}_2\text{'s action)}$$

$$\geq \delta^T c \cdot (1 - o_T(1)) \cdot \mathbb{E}\left[\sum_{t=1}^T \|\mathbf{y}^t - \mathbf{y}(\mathbf{x}^\star; G)\|\right]$$

$$\text{(Successful learning criteria)}$$

$$\mathbb{E}[\text{swap-regret}] \in O(T^a) \qquad\qquad (\pi_2\text{'s swap regret bound})$$

$$\implies \left[\sum_{t=1}^T \|\mathbf{y}^t - \mathbf{y}(\mathbf{x}^\star; G)\|\right] \in O(T^a/\delta^T). \tag{1}$$

$P_1$'s cumulative utility satisfies:

$$\mathbb{E}\left[\sum_{t=1}^T U_1(\mathbf{x}^t, \mathbf{y}^t; G)\right] \geq \sum_{t=1}^T \mathbb{E}[Z^t]\left(U_1(\mathbf{x}'_{\delta^T}, y(\mathbf{x}^\star; G); G) - c_1 \mathbb{E}[\|\mathbf{y}^t - \mathbf{y}(\mathbf{x}^\star; G)\|_1]\right) + c_2 \sum_{t=1}^T \mathbb{E}[1 - Z^t]$$

$$\geq \sum_{t=1}^T \mathbb{E}[Z^t]\left(\text{StackVal}_1(G) - \delta^T c_3 - c_1 \mathbb{E}[\|\mathbf{y}^t - \mathbf{y}(\mathbf{x}^\star; G)\|_1]\right) + c_2 \sum_{t=1}^T \mathbb{E}[1 - Z^t],$$

where

$$c_1 = U_1(\mathbf{x}'_{\delta^T}, \mathbf{y}(\mathbf{x}^\star; G); G) - \min_{y \in \mathcal{A}_2 \setminus \{y(\mathbf{x}^\star; G)\}} U_1(\mathbf{x}'_{\delta^T}, y; G),$$

$$c_2 = \min_{x \in \mathcal{A}_1, y \in \mathcal{A}_2} U_1(x, y),$$

$$c_3 = \text{StackVal}_1(G) - U_1(\bar{\mathbf{x}}, y(\mathbf{x}^\star; G); G).$$

Finally, we use Equation (1) and $\mathbb{E}[Z^t] = 1 - o_t(1)$ to conclude that

$$\mathbb{E}\left[\sum_{t=1}^T U_1(\mathbf{x}^t, y^t)\right] \geq \text{StackVal}_1(G) \cdot T - c_3 \delta^T T - c_1 O(T^a)/\delta^T - o(T). \tag{2}$$

In eq. (2), the second term $c_3 \delta^T T$ comes from playing action $\bar{\mathbf{x}}$ instead of $\mathbf{x}$ to induce a unique best response; the third term $c_1 O(T^a)/\delta^T$ comes from the swap regret of $P_2$, and the last $o(T)$ term comes from errors in the external signals $Z_t$. By choosing $\delta^T = T^{-b}$ for some $0 < b < 1 - a$, $\pi_1$ yields $\text{StackVal}_1(G) \cdot T - o(T)$ utility to $P_1$. In the special case where the last $o(T)$ term is zero (e.g., when $P_1$ has perfect knowledge about the game $G$ as in Theorem 3.1), we can achieve the optimal tradeoff by setting $b$ to be $\frac{1-a}{2}$, which yields a convergence rate of $O(T^{\frac{1+a}{2}})$ to $\text{StackVal}_1(G)$.

The doubling trick to construct $\pi_1$ that does not rely on the time horizon being known: Initialize a maximum horizon $T_m$. Until the round index $t$ hits $T_m$, employ $\mathbf{x}'_{\delta^{T_m}}$. Once $t = T_m$, update $T_m \leftarrow 2T_m$ and employ $\mathbf{x}'_{\delta^{T_m}}$ until the round index exceeds the new max horizon.

**Remark B.1** (Rate of convergence to Stackelberg benchmark.)**.** *Algorithms that provide swap-regret rates of $O(T^{1/2})$ are known [7, 37]. By setting $a = \frac{1}{2}$ and $b = \frac{1-a}{2}$, our proof shows that these algorithms lie in a PNE of the meta-game that results in a $O(T^{3/4})$ convergence rate to $P_1$'s Stackelberg benchmark.*

### B.3 Proof of Theorem 3.2

We prove this theorem for every fixed signal precision $p_2 = p \leq p^\star$. Before diving into the proof, we will first introduce some notations. For $(i, j) \in \{1, 2\} \times \{1, 2\}$, we use $\mathsf{CSP}_{ij}$ to denote the CSP

|   | C | | D | |
|---|---|---|---|---|
| A | $16/\gamma,$ | $1$ | $16/\gamma,$ | $-32/\gamma$ |
| B | $2,$ | $0$ | $0,$ | $2$ |

Game Matrix $G_1$

|   | C | | D | |
|---|---|---|---|---|
| A | $1,$ | $1$ | $0,$ | $-32/\gamma$ |
| B | $0.9,$ | $0$ | $0.1,$ | $2$ |

Game Matrix $G_2$

Figure 4: game matrices $G_1$ and $G_2$. $P_1$ is the row player and $P_2$ is the column player. The values in each cell are ($P_1$'s utility, $P_2$'s utility). Shaded cells represent the action profiles supported in the Stackelberg equilibria led by the column player. The parameter $\gamma$ is defined as $\frac{1-p^\star}{1+p^\star} \in (0, 1]$, where $p^\star$ is the precision threshold of $p_2$.

induced by $\mathcal{T}^T(\pi_1, \pi_2; s_1 = G_i, s_2 = G_j, G = G_i)$. Note that we have taken $s_1$ and $G$ to both be $G_i$ because $s_1 \equiv G$ when $p_1 = 1$. Recall that $\mathcal{T}^T(\pi_1, \pi_2; G_i, G_j, G_i)$ is the distribution over trajectories of length $T$ (denoted as $\tau = (\mathbf{x}^t, \mathbf{y}^t)_{t \in [T]}$) generated by the pair of algorithms $(\pi_1, \pi_2)$ when $\pi_1$ takes input signal $s_1 = G_i$ and $\pi_2$ takes input signal $s_2 = G_j$. Formally, we have

$$\mathsf{CSP}_{ij} \triangleq \mathsf{CSP}_{\mathcal{T}^T(\pi_1, \pi_2; G_i, G_j, G_i)} = \mathop{\mathbb{E}}_{\tau \sim \mathcal{T}^T(\pi_1, \pi_2; G_i, G_j, G_i)} \left[ \frac{1}{T} \sum_{t=1}^{T} \mathbf{x}_t \otimes \mathbf{y}_t \right].$$

In addition, for $i \in \{1, 2\}$, we use $\mathsf{CSP}_i$ to denote the CSP generated by the trajectory distribution $\mathcal{T}^T(\pi_1, \pi_2; G_i)$. Since $s_1$ perfectly correlates with $G$, and $s_2$ is drawn from the signal distribution $\varphi_{p_2}(s | G) = \frac{1+p_2}{2} \cdot \mathbb{1}\{s = G\} + \frac{1-p_2}{2} \cdot \mathbb{1}\{s \neq G\}$, we can equivalently write $\mathsf{CSP}_1$ and $\mathsf{CSP}_2$ as

$$\mathsf{CSP}_1 = \frac{1+p}{2} \mathsf{CSP}_{11} + \frac{1-p}{2} \mathsf{CSP}_{12};$$

$$\mathsf{CSP}_2 = \frac{1-p}{2} \mathsf{CSP}_{21} + \frac{1+p}{2} \mathsf{CSP}_{22}.$$

As argued in the proof sketch, we will establish the following three claims about $\mathsf{CSP}_1$ and $\mathsf{CSP}_2$.

**Claim B.2.** *If $(\pi_1, \pi_2)$ forms an equilibrium in the algorithm space, then $\mathsf{CSP}_1(B, D) \leq \frac{\gamma}{8} + o_T(1)$.*

**Claim B.3.** *If $(\pi_1, \pi_2)$ forms an equilibrium in the algorithm space, then $\mathsf{CSP}_1(A, D) \leq \frac{\gamma}{8} + o_T(1)$.*

**Claim B.4.** *If Claim B.3 holds and $(\pi_1, \pi_2)$ grants $P_2$ the expected Stackelberg value, i.e.,*

$$\liminf_{T \to \infty} \bar{U}_2(\pi_1, \pi_2; \mathcal{D}) \geq \frac{3}{2},$$

*then $\mathsf{CSP}_2(B, D) \geq 1/2 + o(1)$.*

We will first build the proof of Theorem 3.2 on these claims, and then formally establish these claims in appendix B.4.

Assume that $(\pi_1, \pi_2)$ is a PNE in the meta-game that lets $P_2$ achieve the benchmark StackVal$(\mathcal{D}) = \frac{3}{2}$. We show that if $(\pi_1, \pi_2)$ satisfies Claims B.2 through B.4, then $P_1$ gains utility by deviating to an algorithm $\pi_1'$ that always plays according to $G_1$.

Formally, consider $\pi_1'$ defined as follows. For any history time step $t > 0$ and any history $H_1^r$ ($r > 0$),

$$\pi_1'(G, H_1^{1:t-1}) \triangleq \pi_1(G_1, H_1^{1:t-1}), \quad \forall G \in \{G_1, G_2\}.$$

After the deviation, $P_1$'s average utility can be expressed as

$$\bar{U}_1^T(\pi_1', \pi_2) = \frac{1}{2} \bar{U}_1(\pi_1, \pi_2; G_1) + \frac{1-p}{4} \mathop{\mathbb{E}}_{\tau \sim \mathcal{T}^T(\pi_1', \pi_2; G_2, G_1, G_2)} \left[ \frac{1}{T} \sum_{t=1}^{T} U_1(\mathbf{x}_t, \mathbf{y}_t; G_2) \right]$$

$$+ \frac{1+p}{4} \mathop{\mathbb{E}}_{\tau \sim \mathcal{T}^T(\pi_1', \pi_2; G_2, G_2, G_2)} \left[ \frac{1}{T} \sum_{t=1}^{T} U_1(\mathbf{x}_t, \mathbf{y}_t; G_2) \right].$$

Since $\pi_1'$ is defined to behave according to $\pi_1$ on observing $G_1$, the above equation can be rewritten as

$$\bar{U}_1(\pi_1', \pi_2) = \frac{1}{2}\bar{U}_1(\pi_1, \pi_2; G_1) + \frac{1-p}{4} \mathop{\mathbb{E}}_{(x,y)\sim\mathsf{CSP}_{11}} [U_1(x, y; G_2)]$$
$$+ \frac{1+p}{4} \mathop{\mathbb{E}}_{(x,y)\sim\mathsf{CSP}_{12}} [U_1(x, y; G_2)].$$

Now we define a new CSP to be the mixture of $\mathsf{CSP}_{11}$ and $\mathsf{CSP}_{12}$ but $\mathsf{CSP}_{12}$ is taking up more probability mass than $\mathsf{CSP}_{11}$:

$$\mathsf{CSP}_1' \triangleq \frac{1-p}{2}\mathsf{CSP}_{11} + \frac{1+p}{2}\mathsf{CSP}_{12}.$$

Note that compared with the $\mathsf{CSP}_1$ defined before (repeated below)

$$\mathsf{CSP}_1 = \frac{1+p}{2}\mathsf{CSP}_{11} + \frac{1-p}{2}\mathsf{CSP}_{12},$$

we can conclude that for any action pair $(x, y) \in \{A, B\} \times \{C, D\}$,

$$\mathsf{CSP}_1'(x, y) \leq \frac{1+p}{1-p}\mathsf{CSP}_1(x, y) \leq \frac{1+p^\star}{1-p^\star}\mathsf{CSP}_1(x, y) = \frac{1}{\gamma}\mathsf{CSP}_1(x, y),$$

where the second step used $p \leq p^\star$ and the last step uses the definition of $\gamma = \frac{1-p^\star}{1+p^\star}$. Combined with the upper bounds on $\mathsf{CSP}_1(B, D)$ and $\mathsf{CSP}_1(A, D)$ from Claim B.2 and Claim B.3, we have

$$\mathsf{CSP}_1'(B, D) + \mathsf{CSP}_1'(A, D) \leq \frac{1}{\gamma}(\mathsf{CSP}_1(B, D) + \mathsf{CSP}_1(A, D)) \leq \frac{1}{4} + o_T(1). \qquad (3)$$

Finally, we lower bound the increase in $\mathrm{P}_1$'s utility after deviating from $\pi_1$ to $\pi_1'$ as follows. Since

$$\bar{U}_1^T(\pi_1', \pi_2) - \bar{U}_1^T(\pi_1', \pi_2) = \frac{1}{2}\left(\mathop{\mathbb{E}}_{(x,y)\sim\mathsf{CSP}_1'} [U_1(x, y; G_2)] - \mathop{\mathbb{E}}_{(x,y)\sim\mathsf{CSP}_1} [U_1(x, y; G_2)]\right)$$

it suffices to lower bound the difference in utility when $G_2$ is realized.

On the one hand, note that $U_1(x, y; G_2) \geq 0.9$ as long as $(x, y) \neq (A, D)$ or $(B, D)$, we have

$$\mathop{\mathbb{E}}_{(x,y)\sim\mathsf{CSP}_1'} [U_1(x, y; G_2)] \geq (1 - \mathsf{CSP}_1'(A, D) - \mathsf{CSP}_1'(B, D)) \cdot 0.9$$
$$\geq \left(\frac{3}{4} - o_T(1)\right) \cdot 0.9 > 0.6 - o_T(1). \qquad \text{(from eq. (3))}$$

On the other hand, since $U_1(x, y; G_2) \leq 0.1$ when $(x, y) = (B, D)$ and $U_1(x, y; G_2) \leq 1$ otherwise, we have

$$\mathop{\mathbb{E}}_{(x,y)\sim\mathsf{CSP}_1} [U_1(x, y; G_2)] \leq \mathsf{CSP}_1(B, D) \cdot 0.1 + (1 - \mathsf{CSP}_1(B, D)) \cdot 1$$
$$\leq \frac{1}{2} \cdot 1.1 + o_T(1) \qquad (\mathsf{CSP}_1(B, D) \geq \frac{1}{2} + o_T(1) \text{ from Claim B.4})$$
$$< 0.6 + o_T(1).$$

As a result, after taking their difference, we have that in the asymptotic regime

$$\limsup_{T\to\infty} \left(\bar{U}_1^T(\pi_1', \pi_2; \mathcal{D}) - \bar{U}_1^T(\pi_1', \pi_2; \mathcal{D})\right) > 0,$$

which contradicts with the assumption that $(\pi_1, \pi_2)$ forms a PNE (cf. Definition 2.1) in the meta-game! Therefore, it cannot be possible for any PNE $(\pi_1, \pi_2)$ of the meta game to achieve the benchmark $\mathsf{StackVal}_2(\mathcal{D})$ for $\mathrm{P}_2$. The proof is thus complete.

## B.4 Proof of Technical Claims

*Proof of Claim B.2.* We prove this claim by contradiction. Assume that the above claim does not hold, i.e.,

$$\limsup_{T \to \infty} \mathsf{CSP}_1((B, D)) > \frac{\gamma}{8},$$

we will show that $P_1$ gains utility by deviating to another algorithm $\pi_1'$ that always plays action $A$ regardless of the signals and the feedbacks observed. We have

$$\liminf_{T \to \infty} \bar{U}_1^T(\pi_1, \pi_2; \mathcal{D})$$

$$= \liminf_{T \to \infty} \left( \frac{1}{2} \bar{U}_1^T(\pi_1, \pi_2; G_1) + \frac{1}{2} \bar{U}_1^T(\pi_1, \pi_2; G_2) \right)$$

$$= \liminf_{T \to \infty} \left( \frac{1}{2} \mathop{\mathbb{E}}_{(x,y) \sim \mathsf{CSP}_1} U_1(x, y; G_1) + \frac{1}{2} \mathop{\mathbb{E}}_{(x,y) \sim \mathsf{CSP}_2} U_1(x, y; G_2) \right)$$

$$\leq \liminf_{T \to \infty} \left( \frac{1}{2} \left( \mathsf{CSP}_1(B, D) \cdot 0 + (1 - \mathsf{CSP}_1(B, D)) \cdot \frac{16}{\gamma} \right) + \frac{1}{2} \cdot 1 \right)$$
$$\qquad (U_1(x, y; G_1) \leq \frac{16}{\gamma} \text{ for all } (x, y) \neq (B, D); U_2(x, y; G_2) \leq 1 \text{ for all } (x, y))$$

$$< \frac{1}{2} \left( 1 - \frac{\gamma}{8} \right) \frac{16}{\gamma} + \frac{1}{2} \qquad \text{(assumption that } \limsup_{T \to \infty} \mathsf{CSP}_1((B, D)) > \frac{\gamma}{8})$$

$$= \frac{8}{\gamma} - \frac{1}{2}.$$

On the other hand, since $P_1$ always plays action $A$ under $\pi_1'$, which has utility $\frac{16}{\gamma}$ in $G_1$ regardless of the strategy of $P_2$, we have

$$\limsup_{T \to \infty} \bar{U}_1^T(\pi_1', \pi_2; \mathcal{D}) \geq \limsup_{T \to \infty} \frac{1}{2} \bar{U}_1^T(\pi_1', \pi_2; \mathcal{D}) \geq \frac{1}{2} \cdot \frac{16}{\gamma} = \frac{8}{\gamma}.$$

Combining the above two inequalities give us

$$\limsup_{T \to \infty} \left( \bar{U}_1^T(\pi_1', \pi_2; \mathcal{D}) - \bar{U}_1^T(\pi_1, \pi_2; \mathcal{D}) \right) > 0,$$

which violates the requirement of PNE in Definition 2.1. Therefore, we have established the claim that $\mathsf{CSP}_1(B, D) \leq \frac{\gamma}{8} + o_T(1)$. $\qquad \square$

*Proof of Claim B.3.* Again, assume for the sake of contradiction that $\mathsf{CSP}_1(A, D) \leq \frac{\gamma}{8} + o_T(1)$ does not hold, which implies

$$\limsup_{T \to \infty} \mathsf{CSP}_1(A, D) > \frac{\gamma}{8}.$$

We upper bound $P_2$'s utility under equilibrium as

$$\liminf_{T \to \infty} \bar{U}_2^T(\pi_1, \pi_2; \mathcal{D})$$

$$= \liminf_{T \to \infty} \left( \frac{1}{2} \mathop{\mathbb{E}}_{(x,y) \sim \mathsf{CSP}_1} \bar{U}_2(x, y; G_1) + \frac{1}{2} \mathop{\mathbb{E}}_{(x,y) \sim \mathsf{CSP}_2} \bar{U}_2(x, y; G_2) \right)$$

$$\leq \liminf_{T \to \infty} \left( \frac{1}{2} \cdot \mathsf{CSP}_1(A, D) \cdot \left( -\frac{32}{\gamma} \right) + \left( 1 - \frac{1}{2} \cdot \mathsf{CSP}_1(A, D) \right) \cdot 2 \right)$$
$$\qquad (P_2\text{'s utility is } -32/\gamma \text{ for } (A, D) \text{ and } \leq 2 \text{ for all other cells})$$

$$< \frac{\gamma}{16} \cdot \left( -\frac{32}{\gamma} \right) + \left( 1 - \frac{\gamma}{16} \right) \cdot 1 \qquad \text{(assumption that } \limsup_{T \to \infty} \mathsf{CSP}_1(A, D) > \gamma/8.)$$

$$< 0.$$

On the other hand, if $P_2$ deviates to an algorithm $\pi_2'$ that always plays action $C$ regardless of the signal $s_2$ and the observed feedbacks, then the average utility $\bar{U}_2(\pi_1, \pi_2'; \mathcal{D})$ is always nonnegative. As a result, we have

$$\limsup_{T \to \infty} \left( \bar{U}_2^T(\pi_1, \pi_2'; \mathcal{D}) - \bar{U}_2^T(\pi_1, \pi_2, ; \mathcal{D}) \right) > 0,$$

which again violates the condition of $(\pi_1, \pi_2)$ being in a PNE in the meta-game. Therefore, we must have $\mathsf{CSP}_1(A, D) \leq \frac{\gamma}{8} + O_T(1)$. $\qquad \square$

*Proof of Claim B.4.* Note that $U_2$ is the same under $G_1$ and $G_2$, for which the top-2 highest utility values are achieved by $(B, D)$ and $(A, C)$ respectively. Therefore, we can upper bound $\mathrm{P}_2$'s expected average utility as

$$
\begin{aligned}
\bar{U}_2^T(\pi_1, \pi_2; \mathcal{D}) &= \frac{1}{2} \underset{(x,y)\sim\mathsf{CSP}_1}{\mathbb{E}} U_2(x, y; G_1) + \frac{1}{2} \underset{(x,y)\sim\mathsf{CSP}_2}{\mathbb{E}} U_2(x, y; G_2) \\
&\leq \frac{1}{2}\Big( (1 - \mathsf{CSP}_1(B, D)) \cdot 1 + \mathsf{CSP}_1(B, D) \cdot 2 \Big) \\
&\qquad + \frac{1}{2}\Big( (1 - \mathsf{CSP}_2(B, D)) \cdot 1 + \mathsf{CSP}_2(B, D) \cdot 2 \Big) \\
&= 1 + \frac{1}{2}\mathsf{CSP}_1(B, D) + \frac{1}{2}\mathsf{CSP}_2(B, D) \\
&\leq 1 + \frac{\gamma}{16} + \frac{1}{2}\mathsf{CSP}_2(B, D) + o(1),
\end{aligned}
$$

where the last step uses Claim B.3. On the other hand, from the assumption, $\bar{U}_2(\pi_1, \pi_2; \mathcal{D})$ is lower bounded by $3/2$ as $T \to \infty$. Therefore, we obtain

$$
\lim_{T\to\infty} \mathsf{CSP}_2(B, D) \geq 1 - \frac{\gamma}{8} \geq \frac{1}{2},
$$

which proves $\mathsf{CSP}_2(B, D) \geq \frac{1}{2} + o(1)$. $\qquad\square$

# C Implications from prior works

In this section, we provide arguments for the following claims about PNE of the meta-game that are implied either directly or indirectly by previous works. We say that an algorithm $\pi_i$ of $\mathrm{P}_i$ is *supported in a meta-game PNE* if there exists an algorithm $\pi_{-i}$ such that the pair $(\pi_i, \pi_{-i})$ forms a PNE of the meta-game. The claims in this section provide answers to whether common classes of algorithms such as (swap)regret-minimizing, myopically best-responding, playing Stackelberg response are always/sometimes/never supported in a meta-game PNE.

**Claim C.1.** *All no-swap-regret algorithms are supported in a meta-game PNE for all games G. Put another way, being no-swap-regret is a sufficient condition for an algorithm to be supported in some meta-game PNE.*

*Proof.* This is a direct corollary of [17, Theorem 6] which can also be justified by our proof of Theorem 3.1. Consider the pair of algorithms $(\pi_1, \pi_2)$ in which $\pi_1$ plays strategies close to the Stackelberg optimal strategy (via a doubling trick, see Appendix B.2 for the full construction), and $\pi_2$ is a no-swap-regret algorithm. We have proved in Appendix B.2 that $(\pi_1, \pi_2)$ is a PNE in the meta-game because no-swap-regret algorithms are able to cap their opponent's utility at the Stackelberg value. $\qquad\square$

The second claim is stated as Theorem 2 by Brown et al. [10]. Since most game matrices do not have PNE, this claim effectively states that a pair of no-swap-regret algorithms cannot be PNE in the meta-game for most games.

**Claim C.2** (Theorem 2, [10])**.** *Unless the stage game G has a PNE, any pair of two no-swap regret algorithms cannot form a PNE of the meta-game.*

**Claim C.3.** *No-regret is not a sufficient condition for an algorithm to be supported in a meta-game PNE. For common no-regret algorithms such as EXP3, there is a game where no meta-game PNE contains this algorithm.*

*Proof sketch.* This claim can be established by combining two claims in [9]. In their setting, a single seller repeatedly sells a single item to a single buyer for $T$ rounds. We will use two of their results:

1. (Theorem 3.1 of [9]) If the buyer uses EXP3 or other mean-based algorithms, then there exists an algorithm for the seller which extracts (almost) full welfare.

2. (Theorem 3.3 of [9]) There exists an algorithm for the buyer (no-regret without overbidding), which caps the seller's revenue at the Mayerson value.

Wlog, assume $P_1$'s algorithm $\pi_1$ is a mean-based no-regret algorithm such as EXP3. We will show that for any algorithm $\pi_2$ of $P_2$, the pair $(\pi_1, \pi_2)$ cannot be a PNE of the meta-game.

Let $\pi_2'$ be the algorithm given by the above result 1 that lets $P_2$ extract full welfare against $\pi_1$. Let $\pi_1'$ be the algorithm that achieves the property in the above result 2.

On the one hand, if $\bar{U}_1^T(\pi_1, \pi_2) \leq \text{Welfare}(\mathcal{D}) - \Omega(T)$, then $P_1$ will increase utility by deviating to algorithm $\pi_1'$, thus $(\pi_1, \pi_2)$ cannot be a PNE in the meta-game.

On the other hand, if $\pi_1$ is already extracting full welfare against $\pi_2$ (meaning that $P_1$ is getting asymptotically zero utility), then $P_1$ has the incentive to deviate to algorithm $\pi_2'$, under which she can cap $P_2$'s utility at the Myerson value and therefore guarantee herself nonzero utility. For this reason, $(\pi_1, \pi_2)$ cannot be a PNE in the meta-game.

Combining the above two cases, we conclude that $\pi_1$ cannot be supported in any PNE of the meta-game. $\qquad\square$

