# OpenReview forum: "Is Knowledge Power? On the (Im)possibility of Learning from Strategic Interactions"
_NeurIPS.cc/2024/Conference — NeurIPS 2024 poster_

### Official Review · Reviewer_r7Rg · 2024-07-03

**Soundness:** 4
**Presentation:** 4
**Contribution:** 4
**Rating:** 8
**Confidence:** 4

**Summary:**

This paper addresses the theoretical question of whether, through repeated interactions, strategic agents can overcome the uncertainty about the exact payoff structure of the game being played to achieve outcomes they could have achieved in the absence of uncertainty. The authors specifically consider a Stackelberg Game setting, where there are two players, each aiming to maximize their total payoff in the repeated games, with one being more informed about the game structure than the other. The degree of informedness is modeled as a real number $p \in [0,1]$, representing the precision of the signal. It denotes the player's probability of knowing the exact game structure being played each round in addition to a prior distribution. In this setting, the authors study the pure Nash equilibria (PNE) of a meta-game where players choose their decision-making algorithms as their actions. The results demonstrate that when player $P_1$ knows the game perfectly while player $P_2$ does not, there is a clear separation: $P_1$ can always achieve her Stackelberg value, while $P_2$ sometimes cannot. Conversely, if both players are not perfectly certain about the game being played, such separation is provably gone. Overall, this paper advances the theoretical understanding of learning in strategic environments by showing that repeated strategic interactions alone are not enough for an uninformed player to effectively play a Stackelberg Game.

**Strengths:**

1. The topic being studied in this paper is fundamental and relevant to the NeurIPS community.
2. The theoretical findings are clean and fundamental. While non-trivial to prove, the statements of results are concise, and they reveal novel theoretical understandings of learning in strategic environments.
3. The paper is quite well-written, with good typesetting, clear notations, formal statements and proofs, understandable interpretation of results, and proper attribution of them in the introduction.

**Weaknesses:**

I have no major concerns about this work as a theoretical paper in NeurIPS.

If I had to mention a weakness, it would be that the theoretical findings of this work are currently somewhat detached from reality, limiting their direct impact on the real world. That being said, I think this is perfectly fine for a theoretical paper.

**Questions:**

1. Am I correct that each game $G$ in the support of $\mathcal{D}$ must have the same action spaces for both players? Otherwise, it seems that if a player is not perfectly informed about $G$, it is possible for her to make an invalid action.
2. If applicable, can you explain the connection of this work with the real world? Feel free to skip this question if you prefer.

**Limitations:**

I think the authors adequately mentioned the limitations.

---

> ### Author Rebuttal · Authors · 2024-08-07
>
> Thank you for the positive feedback on our work.
>
> > same action spaces for all G in the support of D
>
> Yes, we agree with your point. We do require all the games in the support of D to have the same action spaces for both agents to avoid the issue that you proposed. We will make sure to add this clarification to our model section.
>
> > the connection of this work with the real world
>
> We agree that our work is mainly theoretical. At this stage, our focus is on providing a different lens for the line of work on learning through strategic interactions, rather than immediate applicability to real-world algorithms. However, we believe our work opens the door to many interesting questions about algorithms in the real world. As highlighted in the second paragraph of our discussion section, one interesting open direction is to understand what natural class of algorithms are supported in meta-game PNE, and what benchmarks are achievable when both players are restricted to these classes. In addition, relaxing some modeling assumptions, such as introducing costly signals or considering computationally bounded agents, can further improve the real-world applicability of our work.

---

> > ### Comment · Reviewer_r7Rg · 2024-08-08
> >
> > Thank you for the response. I have read it and decided to stand by my original positive recommendation.

---

### Official Review · Reviewer_LLSw · 2024-07-14

**Soundness:** 3
**Presentation:** 3
**Contribution:** 3
**Rating:** 6
**Confidence:** 3

**Summary:**

In this paper, the authors study whether players can achieve the Stackelberg value when they have uncertainty about the game. Specifically, the authors consider a two-player setting where players can repeatedly interact with the environment. They consider the pure Nash equilibrium in the meta game where the strategies are long-term algorithms. They demonstrate that (1) when one player is perfectly informed, there exists a PNE where the player can achieve his Stackelberg value, (2) when one player is perfectly informed, there is a game where no PNE allows the other player to achieve his Stackelberg value, and (3) when both players are not perfectly informed, both players may not achieve their Stackelberg value in any PNE.

**Strengths:**

1. Studying the results for games with uncertainty is an important and interesting research direction.
2. The theoretical results are generally sound.
3. The paper is well-written and easy to follow.

**Weaknesses:**

I do not have major issues with this paper, but some results need further clarification.
1. In Section 3.2, the authors explain that learning and acting on this learned knowledge are intertwined. I would expect the authors to provide more details. Intuitively, in my opinion, if the less-informed player could estimate the game correctly, he can then behave as the perfectly informed player. Since the authors study the case when $T \rightarrow \infty$, it is possible for the less-informed player to study the game for the first $o(T)$ rounds and then use the same strategy as the perfectly informed player. I expect the authors to explain why this would not work.
2. Proposition 4.2 demonstrates that when both $p_1$ and $p_2$ are smaller than 1, both players cannot achieve the Stackelberg value in any PNE of the game. However, the result is different as long as one of $p_1$ or $p_2$ is 1. The change in the results when $p_1 = 1$ and $p_1 < 1$ is "non-smooth," and I expect the authors to provide further insight about this.
3. I found a typo. There are two words "about" in Line 162.

**Questions:**

See the weakness part.

**Limitations:**

See the weakness part.

---

> ### Author Rebuttal · Authors · 2024-08-07
>
> Thank you for the insightful questions.
>
> > further clarification on section 3.2, why cannot thee less-informed player estimate the game for the first o(T) rounds and then use the same strategy as the perfectly informed player for the rest of the rounds
>
> Thank you for the question. We believe this is an important note, and we will explain this in more detail in the revision. In particular, let us explain why your proposed algorithm for P2 may not be supported in any PNE. If P2 attempts to estimate the game in the first o(T) rounds, she would need to gather information about the game through repeated interactions with the informed player P1. However, if P2’s Stackelberg strategy as a “perfectly informed player” is not favorable to P1, P1 would have an incentives to deviate to an alternative algorithm that does not reveal any game information to P2 during these o(T) rounds. For example, P1’s alternative algorithm could behave exactly the same regardless of the true game being played. As a result, P2 is no longer able to accurately estimate the game in the first o(T) rounds. This is also the intuition behind our proof of Theorem 3.2, especially those sketched in Line 274-286.
>
> As a result, for P2’s algorithm to be in equilibrium, either she cannot successfully estimate the game, or, even if she manages to estimate it the first o(T) rounds, she cannot deploy the Stackelberg strategy of the estimated game in the remaining the rounds, as P1 would then deviate. In other words, P2 either cannot learn the game or, if she does, she cannot use her learned knowledge to achieve her Stackelberg value benchmark.
>
>
> > insights into why the change in results from $p_1=1$ to $p_1<1$ is non-smooth
>
> Intuitively, when $p_1=1$, the player P1 already has perfect knowledge about the game, so there is no additional information for P1 to learn through interactions. As a result, whether P2 deviates or not does not impact P1’s ability to implement his Stackelberg strategy. However, when $p_1$ is even slightly smaller than 1, there remains a small amount of information that P1 needs to learn from interactions with P2. In this case, P2 could potentially have (though tiny) incentive to deviate to an alternative algorithm that prevents P1 from learning this remaining information, similar to the scenario described in the previous question. In other words, this non-smoothness is inherently similar to the non-smoothness of Nash equilibria generally, as even minimal incentive to deviate from a strategy can disqualify a pair of strategies from being an equilibrium.
>
> On a technical level, our lower bound proof (Theorem 3.2) for the case when $p_2\to1$ relies on constructing game matrices with utilities that scale inversely to $1-p_2$. Specifically, this lower bound construction would no longer be valid if $p_2=1$ because the game needs to be finite.

---

> > ### Comment · Reviewer_LLSw · 2024-08-12
> >
> > I appreciate the authors' response and I will maintain my score.

---

### Official Review · Reviewer_bNuD · 2024-07-14

**Soundness:** 3
**Presentation:** 2
**Contribution:** 3
**Rating:** 6
**Confidence:** 3

**Summary:**

The paper studies two-player repeated games where both players use (no-regret) learning algorithms to choose strategies simultaneously in each stage, which are called meta-games. The authors define the pure Nash equilibria of the meta-games and explored the players' equilibrium utilities based on their initial information  about the game. Specifically, they find that if one player is fully informed of the game while the other is partially informed, the fully informed player can always guarantee its Stackelberg utility at some equilibrium, while the partially informed one cannot. When both players are partially informed, then there are cases that either can fail to obtain her Stackelberg value.

**Strengths:**

The meta-games where players use learning algorithms to play against each other is natural and important, due to the widespread use of machine learning techniques. The results into how initial information asymmetries influence the utilities players can achieve in equilibrium is interesting.

**Weaknesses:**

1. The paper can be presented more clearly and rigorously in at least the following ways:
    1. Beginning with the discussion of Stackelberg games in the introduction is confusing as the paper considers repeated simultaneously games.
    2. The reason why take Stackelberg value as a benchmark is lacking.
    3.The statements of the theorems do not mention which learning classes are used (no-regret or no-swap regrets, or hold for both), and whether two players have the same sets of learning algorithms.
    4. The paper does not mention whether any meta-game holds a pure NE.
2. The paper lacks a clear explanation of its technical novelty compared to previous work on strategizing no-regret learners?

**Questions:**

1. See weaknesses 2.
2. Do the agents's set of strategies need to be the same to obtain all the results?
3. Does the meta-game always admit a pure NE?

**Limitations:**

1. The related work with on "Information asymmetry in repeated games" is missing references. I suggest the authors refer to the "related work" section in the paper they cite, "learning to manipulate a commitment optimizer", to include all references. Two references that I know are missing are:

    "
- Thanh H. Nguyen and Haifeng Xu. Imitative attacker deception in Stackelberg security games. IJCAI'19

- Yurong Chen, Xiaotie Deng, and Yuhao Li. Optimal private payoff manipulation against commitment in
extensive-form games. WINE'22
    "

2. typos:
    1. line 338: "that that" -> "that"

---

> ### Author Rebuttal · Authors · 2024-08-07
>
> Thank you for the feedback.
>
> > Reason for using Stackelberg value as a benchmark
>
> Thank you for your question. We agree that more information on choosing Stackelberg value as a benchmark will strengthen the paper. We plan to include more details, as we briefly discuss below.
>
> We view our work as providing a different lens on studying learnability in the presence of strategic interactions that also elucidates the context and subtleties of a vast line of prior work in this space. Therefore, we use the benchmark that is primarily studied in this line of work, which by and large, uses the one-shot game's Stackelberg value as the benchmark for the repeated game. Stackelberg value of the one-shot game forms a particularly compelling benchmark to study because it is both achievable and the tightest under some assumptions in prior work [Brown et al.; Haghtalab et al.; NeurIPS'23].
>
> The Stackelberg value is also unique for general sum games, unlike other equilibria classes (such as NE and CE). Hence, the Stackelberg value provides a clean way to show separation between the benchmarks achievable by informed and less-informed players in the meta-game’s PNE.
>
> > Which learning classes are used?
>
> We believe there may be a misunderstanding about our setting, as described by the reviewer’s summary. We do not restrict either player to specific learning classes, such as no-regret or no-swap regret algorithms. Instead, we allow both players to choose any algorithm from the entire space of all possible algorithms, as long as they form best responses to each other. In Line 161-163 we define algorithms as sequences of mappings from the player’s information about the game and the historical observations to the (randomized) strategies in the next round. We are intentional in this choice because, as this (see Appendix C) and some prior work show, a pair of no-regret algorithms cannot form an equilibrium with each other. Therefore, to model the long-term behaviors of both agents, we need a much larger and more expressive space of algorithms.
>
> > Do the agents' sets of strategies need to be the same?
>
> We interpret the “set of strategies” in your question as referring to the strategies in the meta-game, ie., the long-term strategies or algorithms. Indeed, the set of long-term strategies (i.e., algorithms) is semantically the same for both agents, which is the space of all possible algorithms. In Lines 161-163 of our paper, all algorithms can be written as sequences of mappings from the signal $s_i$ and the historical observations to the strategies that the agents want to use in the next round. The only difference between the two agent’s set of algorithms is that their input signals $s_1$ and $s_2$ may carry different amounts of information.
>
> If your question refers to whether both agents’ sets of pure strategies are the same for all one-shot games G in the support of D, the answer is also yes. This is implicit in our paper, and we will clarify it in future versions.
>
> > Technical novelty compared to previous work on strategizing against no-regret learners
>
> Our main contribution is conceptual, providing a framework for interpreting previous work on strategizing against algorithm classes like no-regret, rather than introducing new tools. Our framework results in the following key takeaways:
>
> **Takeaway 1:** In settings with informational asymmetry, informational advantage can persist throughout repeated interactions. We show this by constructing an instance where no PNE of the meta-game allows the less-informed player to achieve her Stackelberg value. This contrasts previous works that show that it is always possible to learn unknown information and achieve the Stackelberg value when interacting with agents employing specific classes of algorithms. This difference is due to the pair of algorithms considered in previous works not being a PNE of the meta-game and hence having differing levels of rationality.
>
> **Takeaway 2:** The persistence of informational advantage is because learning and acting based on the learned knowledge are intertwined. We argue that this is not due to the less-informed player being unable to learn the game, but because if she uses this learned information to achieve her Stackelberg value, her opponent would benefit from deviating to a different algorithm that does not reveal knowledge of the game to prevent her from learning.
>
> On the technical front, in our lower bound construction showing that no PNE allows the less-informed player to achieve her Stackelberg value (Theorem 3.2), we use the concept of correlated strategy profiles (CSP) introduced by [Arunachaleswaran, 2024] as a simple sufficient statistic for the average utilities resulting fromdue to pairs of arbitrary algorithms. By using CSPs, we can greatly simplify the analysis, reducing the complexity of the problem from high-dimensional distributions over trajectories to the lower-dimensional distributions over stage game strategy pairs. This simplification enables us to conduct a careful case analysis of the lower-dimensional distribution, identifying the conditions and structures of distributions that can correspond to PNEs in the meta game. Our results provide further evidence that such techniques can be valuable for understanding the trajectories of strategic repeated interactions.
>
> > Does the meta-game always admit a PNE?
>
> For the main setting we studied (one player is fully informed), the meta game always admits a PNE. In the proof of Theorem 3.1, we have explicitly constructed a PNE. When both players are partially informed, we suspect that the existence of PNE can be formally established similar to the extensions of the Folk Theorem [Wiseman, Econometrica'05]. We will add a remark to future versions of our paper.
>
> > Missing references
>
> Thank you for pointing out these references. We have already cited the first reference, but we will add the second one and any other missing citations to future versions of our paper.

---

> > ### Comment · Reviewer_bNuD · 2024-08-11
> >
> > Thanks for the responses. It seems that I indeed misunderstood the use of no-regret algorithms defined at the end of section 2.  And the takeaways are quite interesting. Here are my further responses and questions:
> >
> > 1. It is interesting to see that PNE always exists in the settings considered in section 3. I have some questions about the definition of PNE (Definition 2.1). Will the limits always exist? Or should the lim actually be limsup?
> > 2. If I replace the Stackelberg value with some other benchmark value, I can possibly obtain similar separations, right?
> > 3. In Theorem 3.1, the benchmark value used is $StackVal_i(G)$, while in Theorem 3.2 and Proposition 4.2, the value is $StackVal_i(\mathcal{D})$. Could you explain why these two values are different? Can this still be regarded as a separation? Can Theorem 3.2 and Proposition 4.2 hold for $StackVal_i(G)$?
> >
> > Other minor comments:
> > 1. Could you write the conditions of $p_1=1$ and $p_2$s into the statements of Theorem 3.1 and Theorem 3.2, to make the conditions where the theorems hold clearer?
> > 2. I think I found another typo: There are two "is that" in line 359. Should the second one be redundant?

---

> > > ### Author Response · Authors · 2024-08-12
> > >
> > > Thank you for reading our response and for the further questions. We are glad that you find our takeaways interesting.
> > >
> > >
> > > > It is interesting to see that PNE always exists in the settings considered in section 3. I have some questions about the definition of PNE (Definition 2.1). Will the limits always exist? Or should the lim actually be limsup?
> > >
> > > Note that Thm 3.1 shows the existence of meta-game PNE with the current definition involving lim instead of limsup. As you point out, it is true that the limit of average utility might not exist for every algorithm but limsup would exist. So considering this alternate notion of PNE with limsup instead of limits could be more natural. Our results continue to hold with this limsup based defintion with appropriate replacements of limits in the proofs with limsup or liminf. We can address this and use this alternate definition in the revised version.
> > >
> > >
> > > > If I replace the Stackelberg value with some other benchmark value, I can possibly obtain similar separations, right?
> > >
> > > It may be possible to show separation through some other benchmark. However, as highlighted in the previous response, we choose the Stackelberg value as the benchmark for showing separation because of its clear interpretability, close connection to previous works, and its uniqueness for all games.
> > >
> > >
> > > > In Theorem 3.1, the benchmark value used is $\text{StackVal}_i(G)$, while in Theorem 3.2 and Proposition 4.2, the value is $\text{StackVal}_i(D)$. Could you explain why these two values are different? Can this still be regarded as a separation? Can Theorem 3.2 and Proposition 4.2 hold for $\text{StackVal}_i(G)$?
> > >
> > > Thank you for the question. This is a subtle yet important difference. Note that achieving $\text{StackVal}_i(G)$ for all G in the support of D is a **stronger condition**, because it requires the player to achieve the Stackelberg value for every realized game. On the other hand, achieving the average Stackelberg value $\text{StackVal}_i(D)$ is a **weaker condition**, as it only requires achieving the benchmark on average across the distribution, not for every realized game. Importantly, if a player can satisfy the stronger condition ($\text{StackVal}_i(G)$ for all G), they automatically satisfy the weaker condition ($\text{StackVal}_i(D)$), but the reverse is not true.
> > >
> > > Therefore, to establish a clearer separation, we have shown that P1 satisfies the stronger condition by achieving $\text{StackVal}_i(G)$ for all G, whereas P2 cannot even satisfy the weaker condition of achieving the average $\text{StackVal}_i(D)$. This is also explained in the remarks in Lines 223-225 and 231-232: P1 can always achieve $\text{StackVal}_i(G)$ for every realized game G, whereas P2 fails to achieve $\text{StackVal}_i(G)$ for some game G.
> > >
> > >
> > > > other comments
> > >
> > > Thank you for the suggestions. We will revise the paper accordingly.

---

> > > > ### Comment · Reviewer_bNuD · 2024-08-13
> > > >
> > > > Thanks for addressing all my questions and comments. I would like to raise my score.

---

### Official Review · Reviewer_CgaS · 2024-07-15

**Soundness:** 4
**Presentation:** 3
**Contribution:** 3
**Rating:** 6
**Confidence:** 4

**Summary:**

The paper explores the impact of information asymmetry on the ability of agents to achieve their Stackelberg optimal strategy in repeated games. It investigates whether agents can overcome initial uncertainty through strategic interactions alone. The authors propose a meta-game model where players' actions are algorithms that determine their strategies based on observed histories and knowledge of the game. The paper's main findings suggest that while an informed player can always achieve her Stackelberg value, an uninformed player cannot necessarily do so, even with repeated interactions.

**Strengths:**

1 The paper presents a clear and novel perspective on information asymmetry in strategic interactions.
2 The theoretical model and meta-game framework are well-defined and contribute to the understanding of learning in games.
3 The analysis of pure Nash equilibria provides valuable insights into the limitations of learning through repeated interactions.

**Weaknesses:**

While the paper discusses the inability of uninformed players to achieve their Stackelberg values, it could provide more insight into the learning dynamics and the rate at which players converge (or fail to converge) to these values.

The findings of the paper, while theoretically sound, may not offer surprising or counterintuitive insights that significantly advance the field.

The authors does not clearly articulate how its findings contribute to the existing body of work in the field of game theory and strategic interactions. The paper would benefit from a clearer exposition of how its findings contribute to the existing body of work.

**Questions:**

Could the authors highlight their contributions and discuss the significance of the results?

**Limitations:**

I did not find any limitations of the paper.

---

> ### Author Rebuttal · Authors · 2024-08-07
>
> Thank you for your comments and questions.
>
> > While the paper discusses the inability of uninformed players to achieve their Stackelberg values, it could provide more insight into the learning dynamics and the rate at which players converge (or fail to converge) to these values.
>
> In our paper, we study the interactions between the informed and uninformed players across a wide range of algorithms, rather than analyzing a specific or parameterized class of algorithms. Because we consider such a broad set of algorithms, we treat the agents’ choices of algorithms as black-boxes. As a result, our primary focus is on determining whether certain benchmarks can be achieved at all, rather than on the learning dynamics or convergence rates.
>
> Nonetheless, when it comes to specific types of interactions that our work addresses, we can provide convergence guarantees using the techniques and results from past work. For example, in demonstrating that the more informed agent can always achieve their Stackelberg value at some PNE (Theorem 3.1), we constructed a pair of algorithms where the informed player (P1) plays strategies close to the Stackelberg strategy and the less informed player (P2) uses a no swap regret algorithm. Implicitly in our proof (see line 627), we show that P1’s convergence rate depends on P2’s swap regret.  For example, when P2’s swap regret is of order $\sqrt{T}$, P1’s convergence rate is $O(T^{3/4})$. On the other hand, our negative results show that in no PNE a sublinear convergence to P2’s Stackelberg value is possible.
>
> We will clarify this point further in the revision by adding a remark about the convergence rate after Theorem 3.1.
>
> > Main contribution, significance of results, and relation to previous work.
>
> We view our work as providing a different lens on studying learnability in the presence of strategic interactions that also elucidate the context and subtleties of a vast line of prior work in this space. By and large, prior work in this space has attempted to establish the following message: “An uninformed player can always learn to achieve/surpass its Stackelberg value through repeated strategic interactions alone.” At a high level, our work demonstrates the opposite, that “In some cases, an uninformed player cannot learn, through repeated interactions alone, to achieve its Stackelberg value”. Of course, these messages, while both technically sound, are contrary to each other. So, what accounts for this difference?
>
> Our work elucidates that the results of prior work (i.e., that learning does happen) hinge on an asymmetry in the rationality levels of the two agents that interact with each other. That is, the dynamics that are studied in prior work involve pairs of agent algorithms that are not best-response to each other. This lack of rationality confounds the takeaway message of prior work, leaving one to wonder whether it was the lack of rationality of the second agent or some genius of the first agent’s algorithm that enabled the first player to learn from strategic interactions.
>
> The starting point of our work is therefore to consider agents that are rational in their choice of algorithm, which we capture by studying the pure Nash Equilibria (PNEs) of the meta-game that is the repeated interactions between two agents. Through this lens, we show that an uninformed agent may not be able to learn to achieve her Stackelberg value (Theorem 3.2), which is what she could have achieved if she was fully informed (Theorem 3.1).
>
> Let us expand on the above by discussing our paper’s takeaways:
>
> **Takeaway 1:** In settings with informational asymmetry, informational advantage can persist throughout repeated interactions. We show this by constructing an instance where no PNE of the meta-game allows the less-informed player to achieve her Stackelberg value. This stands in contrast to results from previous work that show that it is always possible to learn unknown information and achieve the Stackelberg value when interacting with agents from specific classes of algorithms. This difference is due to the pair of algorithms considered in previous work not being a PNE of the meta-game, that is, at least one of the agents was not rational in her choice of the algorithm.
>
> Compared to prior work that also argues that the more-informed player can prevent the less-informed player from realizing her Stackelberg value by misrepresenting her private information, we show this in a stronger sense, without making behavioral assumptions on the more-informed player, and instead analyzing the PNE of the meta-game. Therefore, we show that it is the inherent nature of information asymmetry and not any limit on agent rationality that drives the persistence of the informational advantage throughout repeated interactions.
>
> **Takeaway 2:** The persistence of informational advantage is due to the processes of learning and acting based on the learned knowledge being intertwined. This demonstrates the nuances of why informational advantage persists. We argue that this is not due to the less-informed player being unable to learn the unknown information alone. In fact, in our construction in Theorem 3.2 where the less-informed player is unable to achieve her Stackelberg value, she still does fully learn the game matrix. However, if she deviated to using this learned information to obtain her Stackelberg value, the other player would benefit from deviating to a different algorithm that does not depend on informed player’s knowledge and hence prevents learning the game.
>
> In this work we used the meta-game’s PNE as a way to reflect on the interpretations of previous work on repeated strategic interactions. We hope this framework will be useful for future work to also shed light on natural algorithms that may be used and how this depends on the structure of the game and the forms of information available to each agent.

---

> > ### Comment · Reviewer_CgaS · 2024-08-13
> >
> > Thank you for your response. I will maintain my score.

---

### Decision · Program_Chairs · 2024-09-25

**Decision:**

Accept (poster)

**Comment:**

The paper studies how agents can overcome uncertainty about their utilities through repeated interaction in strategic environments. The main focus of the paper is on the impact of information asymmetry on the ability of agents of reaching their Stackelberg value.

All the Reviewers agree that the problem studied in the paper is interesting and relevant to the NeurIPS community. In particular, the idea of meta-game introduced in this paper to study the dynamics generated by two learning algorithms repeatedly interacting in a strategic environment is novel and natural. Thus, **I recommend the paper to be accepted at NeurIPS.**

There are some minor issues raised by the Reviewers that the Authors are strongly encouraged to fix in the final version of the paper. Specifically:

1. The reason why the paper considers the Stackelberg value as a benchmark is lacking, additional discussion on this choice should be added to the paper.

2. While the paper discusses the inability of uninformed players to achieve their Stackelberg values, it could provide more insight into the learning dynamics and the rate at which players converge (or fail to converge) to these values.